# Cochlear transcript diversity and its role in auditory functions implied by an otoferlin short isoform

Huihui Liu[1,2,3,6], Hongchao Liu[1,2,3,6], Longhao Wang[1,2,3,6], Lei Song[1,2,3,6], Guixian Jiang[1,2,3,6], Qing Lu[1,2,3,4], Tao Yang[1,2,3], Hu Peng[5], Ruijie Cai[1,2,3], Xingle Zhao ®[1,2,3], Ting Zhao[1,2,3] & Hao Wu ®[1,2,3] ✉

Isoforms of a gene may contribute to diverse biological functions. In the cochlea, the repertoire of alternative isoforms remains unexplored. We integrated single-cell short-read and long-read RNA sequencing techniques and identified 236,012 transcripts, 126,612 of which were unannotated in the GENCODE database. Then we analyzed and verified the unannotated transcripts using RNA-seq, RT-PCR, Sanger sequencing, and MS-based proteomics approaches. To illustrate the importance of identifying spliced isoforms, we investigated otoferlin, a key protein involved in synaptic transmission in inner hair cells (IHCs). Upon deletion of the canonical otoferlin isoform, the identified short isoform is able to support normal hearing thresholds but with reduced sustained exocytosis of IHCs, and further revealed otoferlin functions in endocytic membrane retrieval that was not well-addressed previously. Furthermore, we found that otoferlin isoforms are associated with IHC functions and auditory phenotypes. This work expands our mechanistic understanding of auditory functions at the level of isoform resolution.

RNA expression at the isoform level is biologically more informative than at the gene level, and different isoforms of the same gene can have different biological functions. Mammalian genes typically produce multiple transcripts[1,2], contributing to their transcriptomic and proteomic diversity relative to other organisms. Alternative splicing (AS) plays an instrumental role in cell differentiation, development[3], tissue-identity acquisition, and maintenance of cell-type-specific properties[4,5]. It is estimated that more than 60% of multiexonic genes in mice undergo AS, which has the potential to create multiple functional protein isoforms from a single gene locus[2]. The current mouse genome annotations are primarily based on the results of short-read RNA sequencing from nonauditory tissues[5,6]. As such, the splicing isoforms in the inner ear remain poorly characterized. It is conceivable that the combined existence of multiple AS isoforms and their dynamic change over time, as well as environmental stress, could drive underlying pathophysiological mechanisms that have been overlooked mainly due to technical limitations regarding measurement of cells in the inner ear. Thus, the effective identification of these isoforms would provide valuable clues for basic and clinical hearing research.

Although the complexity of AS has been well-studied in the central nervous system, very few genes, such as *Myo7a*, *Xirp2*, *Pcdh15*, *Ush1c*, and *Whrn*, have been identified to have alternative isoforms in the inner ear[7–14]. Alternative splicing can contribute to unexpected findings in auditory processing. For example, Whirlin has two major

[1]Department of Otolaryngology-Head and Neck Surgery, Shanghai Ninth People's Hospital, Shanghai Jiao Tong University School of Medicine, Shanghai 200011, China. [2]Ear Institute, Shanghai Jiao Tong University School of Medicine, Shanghai 200011, China. [3]Shanghai Key Laboratory of Translational Medicine on Ear and Nose Diseases, Shanghai 200011, China. [4]Key Laboratory for the Genetics of Developmental and Neuropsychiatric Disorders, Ministry of Education, Bio-X Institutes, Shanghai Jiao Tong University, Shanghai 200240, China. [5]Department of Otolaryngology-Head and Neck Surgery, Changzheng Hospital, Second Military Medical University, Shanghai 200003, China. [6]These authors contributed equally: Huihui Liu, Hongchao Liu, Longhao Wang, Lei Song, Guixian Jiang. ✉e-mail: wuhao@sh9hospital.org.cn

splice variants, a long isoform (Whrn-L) and a short isoform (Whrn-S), which show differential expression in the stereocilia of inner and outer hair cells. Differential isoform expression determines the severity of auditory phenotypes. Mice lacking Whrn-S have been shown to have a moderate auditory phenotype due to abnormal stereocilia morphology, organization, and function in outer hair cells (OHCs), while they have normal stereocilia in inner hair cells (IHCs). In contrast, mice lacking both Whrn-L and Whrn-S develop profound hearing loss due to abnormally short stereocilia in both IHCs and OHCs[11]. Myo7a is another relevant example. In addition to the differential expression of Myo7a isoforms between IHCs and OHCs, the canonical isoform of Myo7a also is expressed in a longitudinal gradient in OHCs that decreases from the apex to the base of the cochlea[13]. It is conceivable that genes expressed in the inner ear might generate distinctive isoform profiles and isoform usage, contributing to the diversity of auditory functions. However, these surveys are mainly based on microarray, PCR, and short-read RNA-seq data, which raises the possibility of underestimating the true complexity of AS events in these gene loci.

Traditional short-read sequencers are not able to capture entire transcripts from end-to-end, and computational assembly based on short-read data makes it challenging to dissect complex transcript isoforms[15]. Thus, the transcriptome repertoire of the cochlea has not been comprehensively studied by short-read (100–400 bp) RNA-seq technologies, as most mammalian mRNA transcripts are 1000–2000 bp in length[16,17]. Long-read sequencing technologies such as Pacific Biosciences (PacBio) show great potential in capturing the true complexity of spliced isoforms. With an average read length of 13.5 kilobases, PacBio allows for the direct reading of full-length transcripts with high accuracy[18]. It has been widely applied to full-length transcriptome sequencing in bulk tissues, cell lines, and sorted cells in multiple organisms[9,19–22]. However, the cochlea is a complex organ, with an estimated 41,500 cells with at least 16 cell subtypes[23]. This limits the yield per cell subtype and again makes it challenging to obtain full-length transcriptional information for each cell-type from bulk tissues. To address these issues, we integrated short-read (Illumina) and long-read (PacBio) sequencing techniques, single-cell isoform RNA sequencing (ScISOr-Seq)[9], to profile the full-length transcriptome at single-cell resolution in the cochlea.

In this study, we identified 126,612 novel transcripts (compared to the GENCODE VM23 database) and verified at least 57,366 (45.31%) using short-read sequencing of the organ of Corti. In addition, we observed that 90.14% of the total novel transcripts had coding capability. Their gene products were verified using mass spectrometry (MS)-based proteomics, confirming the complexity and diversity of spliced isoforms in the cochlea. To investigate the roles of splicing isoforms in auditory functions, we analyzed one gene, *otoferlin* (*Otof*), which was found to generate an inner ear-specific transcript. Otoferlin is a critical protein involved in synaptic vesicle release at the ribbon synapses of IHCs[24,25]. It was chosen because its functional characterization can be precisely measured by electrophysiological approaches under both in vitro and in vivo conditions. We generated an otoferlin canonical isoform (*Otof-C*) deletion (*Otof-ΔC*) animal model, revealing and demonstrated that this novel transcript could be translated into a functional protein isoform that functionally overlaps with the canonical one with decreased vesicle recycling. In addition, compared with the canonical isoform, the novel otoferlin isoform should exhibited decreased binding potential with synaptic proteins such as endophilin A1. These findings will shed new light on otoferlin function in endocytic membrane retrieval, which previously had not been well addressed. It further unveils an important mechanism of hidden hearing loss (HHL) due to impaired synaptic vesicle recycling. It suggests that the proportion of otoferlin isoforms may be tightly associated with the exocytic capacity of IHCs governing the tonotopic gradients of synaptic properties and noise-induced/age-related cochlear synaptopathy. Overall, these results highlight the importance of obtaining a comprehensive view of the full-length transcriptomic of the cochlea, which will expand our mechanistic understanding of auditory functions at the isoform level.

## Results

### Application of a hybrid-sequencing workflow (ScISOr-Seq) to the cells of the cochlea

scRNA-seq (10X Genomics) using an Illumina platform and long-read isoform sequencing (Iso-seq, Pacific Biosciences) were integrated to comprehensively profile the full-length transcriptome of the organ of Corti (Fig. 1a). In our scRNA-seq data, we obtained 9845 cells that passed quality control, with mean gene number of 1829 per cell. Unsupervised clustering and uniform manifold approximation and projection (UMAP) visualization of this dataset reveal 12 distinct clusters with discrete patterns of gene expression (Fig. 1b). To further determine the characteristic expression pattern of each cell type, we screened for differentially expressed genes (DEGs) among the cell clusters and then reclustered all of these DEGs according to their cell-type-specific expression patterns (Fig. 1d).

According to previous studies[26,27], we annotated cell types based on the known cell-type-specific markers (Supplementary Fig. 1a), including Tympanic border cell (TBC, *Emilin2*[+])[28], Fibroblast cell (FC, *Coch*[+], *Otos*[+])[29,30], Inner phalangeal cell/Inner border cell (IPhC/IBC, *Slc1a3*[+], *S100a6*[+])[31–33], Deiter cell/Pillar cell/Inner sulcus cell/Outer sulcus cell (DC/PC/ ISC/OSC, *Gata3*[+], *Fbxo2*[+], *Skp1a*[+], *Epyc*[+], *Bmp4*[+])[26,34–37], Reissner's membrane cell (RMC, *Vom1*[+])[38], Capillary endothelial cells (CEC, *Ly6c1*[+])[39], Capillary pericytes (*Rgs5*[+])[30], OHC (*Ocm*[+], *Slc26a5*[+])[40,41], and IHC (*Otof*[+], *Slc17a8*[+])[25,42]. The marker genes expressed in each cluster are listed in Supplementary Data 1.

For Iso-seq, a total of 5,689,062 full-length circular consensus sequences (CCSs) were detected to be full-length transcripts (with 5′ primers, 3′ primers and poly(A) tails), 5,689,037 of which were tagged with cell barcodes and unique molecular identifier (UMI) sequences. We generated 4,960,442 unique full-length nonchimeric reads (FLNCs) after refinement and deduplication, 99.89% of which could be aligned to the GENCODE reference genome (VM23). Eight cell clusters were identified from our PacBio sequencing data analysis (Fig. 1c, e) and were annotated to specific cell types based on the DEGs (Supplementary Data 2) according to the known markers (Supplementary Fig. 1b). Next, we applied the integration analysis[43] using both scRNA-seq and Iso-seq data to validate the accuracy of our sequence data and found good cell type alignment between cells with shared gene expression patterns for both approaches (Fig. 1f). In addition, we found a strong correlation between Illumina short-read and PacBio long-read data in terms of UMIs and gene counts by cell barcodes (Supplementary Fig. 1c).

Regarding the long-read data analysis, a key challenge was the relatively low sequencing depth of gene number per cell (Supplementary Fig. 1c), which made it difficult to correctly identify cell subtypes, such as IPhC/IBC (Fig. 1c). Short-read 3′ sequencing provided molecular counts for each gene and cell, enabling cell clustering and cell-type assignment using cell-type-specific markers. Single-cell barcodes were also presented in long reads and were used to determine the individual cell of origin for each long read. Therefore, we used short-read sequencing to profile cellular diversity and cell barcodes as a guide to allocate long reads to each cell cluster.

### ScISOr-Seq revealed spliced isoform diversity in the cochlea

Following SQANTI classification, the nonredundant transcripts were classified into four groups (Fig. 2a). These include: i) full splice match (FSM); ii) incomplete splice match (ISM); iii) novel in catalog (NIC); and iv) novel not in catalog (NNC). We excluded several categories (antisense, fusion, genic, genic_intron, and intergenic) as they may be generated from experimental or technical artifacts[44]. In total, we detected 236,012 transcripts from annotated genes in the cochlea. A total of 109,400 (46.35%) of the transcripts were aligned to known

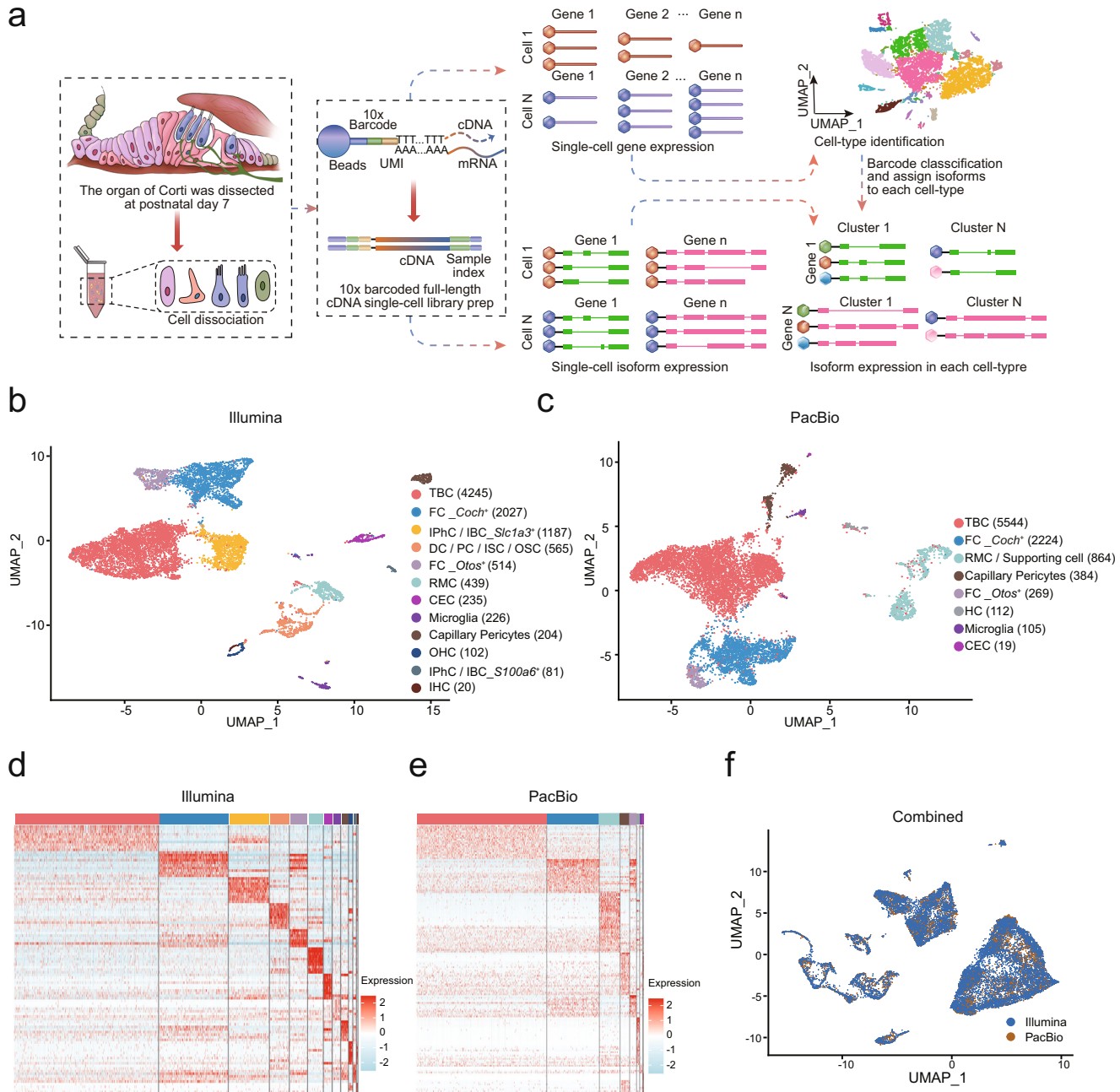

**Fig. 1 | Integration of short- and long-read data for profiling cell-type transcript isoforms. a** Workflow for collecting single-cell full-length isoform sequences from the P7 organ of Corti. **b**, **c** Uniform manifold approximation and projection (UMAP) plot of single cells colored by cell types from the two sequencing platforms, respectively. TBC Tympanic border cell, FC Fibroblast cell, IPhC/IBC Inner pha-langeal cell/Inner border cell, DC/PC/ISC/OSC Deiter cell/Pillar cell/Inner sulcus cell/Outer sulcus cell, RMC Reissner's membrane cell, CEC Capillary endothelial cells, OHC Outer hair cell, IHC Inner hair cell. Cell numbers are shown in brackets. **d**, **e** Heatmap showing the differentially expressed genes (DEGs) between short-read and long-read sequencing identified clusters. The legends indicate the cell type name and the relative expression levels of DEGs are color-coded. **f** UMAP visualization of scRNA-seq and Iso-seq cells following multimodal integration.

reference isoforms (FSM, ISM), and 126,612 (53.65%) were novel tran-scripts (NIC, NNC) (Fig. 2b, Supplementary Data 3), with average read lengths of 1879, 1531, 1992, and 1825 nucleotides for FSM, ISM, NIC, and NNC transcripts, respectively. We also found that 85.08% of the annotated genes (15,262) identified from the long-read sequencing data were able to produce novel transcripts (Fig. 2b). Next, we sum-marized the number of novel transcript isoforms per gene from the Iso-seq reads. Out of 12,986 genes that could produce novel tran-scripts, most of them (83.03%, 10,782) produced multiple novel tran-scripts, far exceeding the numbers previously found in other tissues[9] (Fig. 2c, Supplementary Data 3).

Using ScISOr-Seq data, we assigned and characterized cell cluster-specific expressions for each transcript category. The proportions of each transcript category varied in different cell types, and novel iso-form numbers ranged from 30.08% (IPhC/IBC_*S100a6*+) to 48.89% (TBC) (Fig. 2d, Supplementary Data 4). To determine whether the expression of novel transcripts was cell-specific, an UpSet plot was generated to visualize the intersecting sets from different cell clusters, with a total of 99,692 out of 126,615 novel transcripts showing cell-type specificity (Fig. 2e).

As an example, we selected *Actb*, a ubiquitous gene expressed in all clusters (Fig. 2f), demonstrating the diversity of transcripts in each

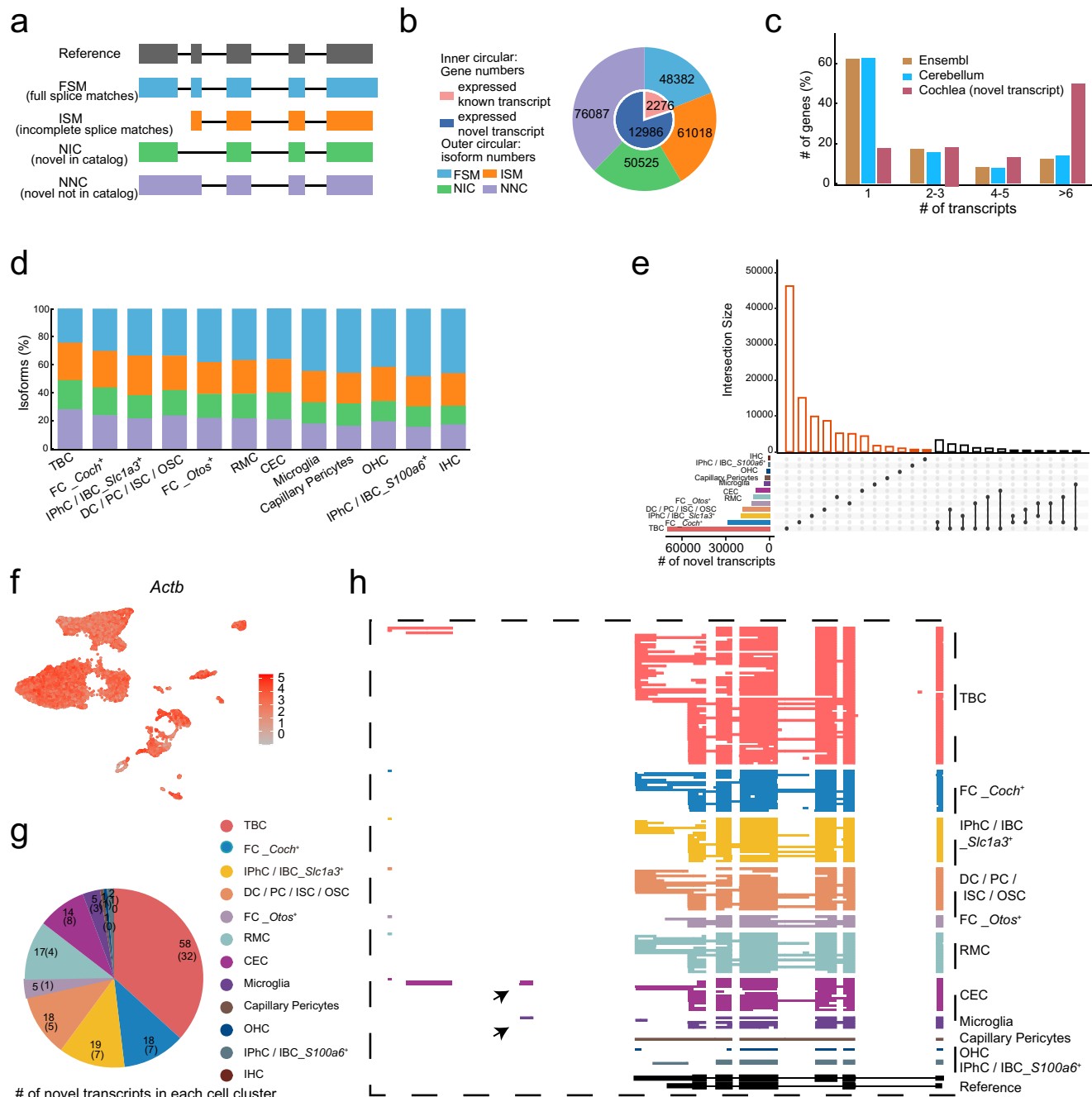

**Fig. 2 | Diverse transcripts in the cochlea. a** The identified isoforms were categorized as follows: FSM (full splice match), ISM (incomplete splice match), NIC (novel in catalog), and NINC (novel not in catalog). **b** Pie charts showing the number of different categories of isoforms and genes from ScISOr-Seq. **c** The number of novel isoforms detected per gene by ScISOr-Seq showed higher isoform diversity than that in the mouse cerebellum and that determined by Ensembl annotation. **d** Distribution of the identified transcript isoform categories across cochlea cell types. **e** UpSet plot comparing the shared novel transcripts among different cell types. Most of the novel transcripts exhibited cell-type specificity. **f** *Actb* expression level among the different cell types, measured by short-read sequencing (cells with high expression are in red). **g** The number of novel transcripts detected by long-read sequencing (the number in the brackets refers to the cell-type specificity transcript number). **h** *Actb* transcripts were identified by ScISOr-Seq and revealed high diversity and cell-type specificity. Each colored row represents the identified transcripts according to each cell type. Blocks represent exons, while the white space represents intronic space. The black track shows the GENCODE annotation of *Actb* (bottom). Arrow: a newly identified exon.

cell type (Fig. 2g, h). From short-read sequencing data, the expression level of *Actb* was indistinguishable from one cell cluster to another. However, within transcript categories, cell specificity was quite distinct (Fig. 2g). That is, among the 100 types of novel transcripts (Supplementary Data 3), 69 were different among cell types (Fig. 2g). Next, we sought to illustrate the presence of diverse transcripts of *Actb* (Fig. 2h), which have 2 transcripts referenced in the GENCODE database (VM23). An annotated exon was found to be preferentially expressed in

nonsensory cells (CEC and Microglia) but was absent in other cell types, highlighting cell-type-specificity.

## Characterization and validation of the AS events of long-read sequencing data

To further characterize the long-read sequencing data, we categorized known and novel transcripts into different types of AS events. These included alternative 5′/3′-donor or acceptor (A5/3), alternative first

exon (AF), alternative last exon (AL), exon skipping (ES), retained intron (RI), and mutually exclusive exon (MX) events (Supplementary Fig. 2a). In total, 231,800 AS events were identified in novel transcripts from 9840 gene loci, including 64,146 ES, 50,655 A3, 35,898 RI, 21,880 AF, 44,855 A5, 7864 MX, and 6502 AL events. Compared to the known transcripts, ES and A3 appear more frequently in novel transcripts (Supplementary Fig. 2b). The distribution of the seven types of AS events shows that a majority of novel transcripts were involved in more than one AS event (Supplementary Fig. 2c), demonstrating the complexity of the inner ear transcriptome.

To validate the spliced transcripts, we used bulk RNA-seq data to verify splicing junctions[45]. We found that the majority (98.87%) of splicing junction motifs from the known transcript could be supported by RNA-seq short reads (Fig. 3a). In addition, among the different types of transcripts, FSM events had the highest verification rates (all junctions were verified with one transcript), indicating the high accuracy of Iso-seq (Fig. 3b). For the novel spliced transcripts, a total of 57,366 (45.31%) transcripts were verified, indicating the novel transcripts were actually transcribed (Fig. 3a, b). Furthermore, we analyzed the proportion of verified spliced junctions of novel transcripts in each cell subtype and found a range of variations. For example, the highest verification rate was 57.84% in inner hair cells and the lowest was 49.57% in Tympanic border cells (Fig. 3c). Together, these analyses strongly support the validity of full-length sequences for various spliced transcripts.

For further validation, we selected six genes that were known to influence hearing using reverse transcription polymerase chain reaction (RT–PCR) and Sanger sequencing (Fig. 3d, e, Supplementary Data 5). The results demonstrated that the sizes of each amplified fragment, as shown by gel banding patterns, were consistent with that of the predicted fragment lengths. These amplified fragments were then cloned for Sanger sequencing. Based on Iso-seq data, a positive correlation was found between cloned fragments and the sequence predictions. In *Gjb6*-PB.10102.9, two PCR bands of 193 bp and 648 bp were observed due to ES events. New exons and intron retention were verified in *S1pr2*, *Calb1*, *Gjb6*, and *Slc26a4* in different cell types. The sequences of these isoforms and their alignments with the corresponding genes are shown in Supplementary Data 5. Of interest, we found that spliced transcript was often verified by Sanger sequencing, but not supported by RNA-seq, such as *Calb1* (PB.21971.7). This suggests that RNA-seq may underestimate the number of verified transcripts. Overall, the novel isoforms really exist.

## Novel transcript isoforms lead to diverse alternative protein products in the cochlea

To determine whether diverse transcripts corresponded to diverse protein products, we analyzed the open reading frames (ORFs, i.e., coding sequences, length ≥100 codons)[46] predicted from the relevant identified transcripts. A majority of FSM (90.58%), ISM (87.82%), NIC (92.14%), and NNC (88.76%) transcripts were predicted to have coding sequences (CDSs) (Supplementary Fig. 2d). Thus, transcript abundance may represent a potential source of proteomic diversity in the inner ear. Since different RNA transcripts may contain identical CDS, we quantified 100,632 unique CDSs from FSM, NIC, and NNC transcripts. We found that 81.90% of the total genes that transcribing novel transcripts had multiple CDSs (Supplementary Fig. 2e). Furthermore, we characterized different types of ORFs as previously described, to further illustrate how transcripts produce diverse proteins[10]. Only 12.49% of the total predicted ORFs were annotated, whereas the majority of ORF types were ORF variants (83.86%) (Supplementary Fig. 2f).

To determine whether the ScISOr-Seq isoforms described here translated into proteins, we extracted protein samples from the organ of Corti and identified them using mass spectrometry (MS)-based proteomics. A total of 89,829 unique peptides that matched the ORF database predicted from ScISOr-Seq data were identified (Supplementary Data 6). A total of 54,541 transcripts were supported by MS-based proteomic data, among which 23,261, 5,724, 8,134, and 17,422 were FSM, ISM, NIC, and NNC isoforms, respectively (Fig. 3f). In addition, we assigned and characterized detected peptides in each cell cluster and observed that the number of transcripts supported by MS-based proteomic datasets ranged from 2,264 (IHC) to 39,967 (TBC) (Supplementary Fig. 2g). In addition, some peptides assigned to each cell cluster exhibited cell-type specific expression patterns (Fig. 3g, Supplementary Data 6).

We discovered 81 ScISOr-Seq specific peptides that were absent from the UniProt database (Supplementary Data 7). This suggests that at least some of the proteins predicted by ScISOr-Seq data are expressed in the inner ear, leading to diverse proteoforms. To reveal the unannotated peptides encoded from these novel transcripts, we selected a deafness-associated gene, *Coch*, as an example. This novel transcript is characterized by exon 5 (63 bp) skipping, which results in a 21 amino acid (AA) deletion and two AA changes (Fig. 3h). This peptide was considered to be from isoform-specific regions since regional mapping could be unequivocally assigned to a unique isoform. Lastly, we observed that these ScISOr-Seq specific peptides were capable of confirming alternative splicing events at the protein level, implying strong evidence for protein diversity in the cochlea.

## Deafness genes exhibit high isoform diversity, and a *Otof* functional short isoform is identified in IHCs

To better understand the transcript diversity of deafness genes in the cochlea, we investigated their expression patterns at full-length resolution. Hearing loss-associated genes such as *Col2a1* also exhibited high isoform diversity, which represented the largest number (210) of novel transcript categories (Supplementary Fig. 3a), despite the fact that it is currently believed to have only two transcripts according to the GENCODE database. In addition, these deafness-related genes also exhibited an increased isoform diversity compared to other genes in the cochlea (Supplementary Fig. 3b).

Among the novel transcripts in deafness-related gene loci, we found that otoferlin, one of the key ribbon synapse proteins, expressed a short transcript originating from an unannotated exon 6b (Supplementary Fig. 3c). It was validated by RT–PCR, Sanger sequencing (Supplementary Fig. 3d; Supplementary Data 5), and long-read sequencing on the ONT platform. Among the tissues expressing *Otof*, such as the cerebral cortex, this short isoform showed inner ear -specificity (Supplementary Fig. 3c, d). It was also found to be highly conserved among mammals (Supplementary Fig. 4a), suggesting its functional importance. Otoferlin is a multi-$C_2$ domain ($C_2$A-F; Fig. 4a) protein[47]. The short isoform is predicted to produce a short-form protein ($C_2$B-F; Fig. 4b) with truncation of the first 192 N-terminal amino acids compared to the canonical protein.

Sound-initiated inner ear Organ of Corti vibration eventually causes deflection of hair bundles of hair cells, which in turn transduce mechanical vibration into changes in receptor potential. The inner hair cell (IHC) is the true sensory cell that in response to receptor potential, releases neurotransmitters to the auditory nerve fibers that innervate the basal lateral membrane. As a calcium sensor, otoferlin is the key molecule mediating vesicle release at the ribbon synapses[47,48]. Absence of otoferlin has been shown to be the primary cause of auditory neuropathy manifested by the loss of synchronized firing of auditory nerve fibers, leading to profound deafness[24,25]. Synchronized firing of auditory nerve fibers can be measured by determining the auditory brainstem response (ABR), whereas neurotransmitter release can be quantified by measuring changes in the membrane capacitance using a whole-cell patch-clamp set up.

We investigated the functional consequences of spliced isoforms to explore the molecular mechanisms driving the functional complexity of auditory processes. Two experimental knockout cohorts

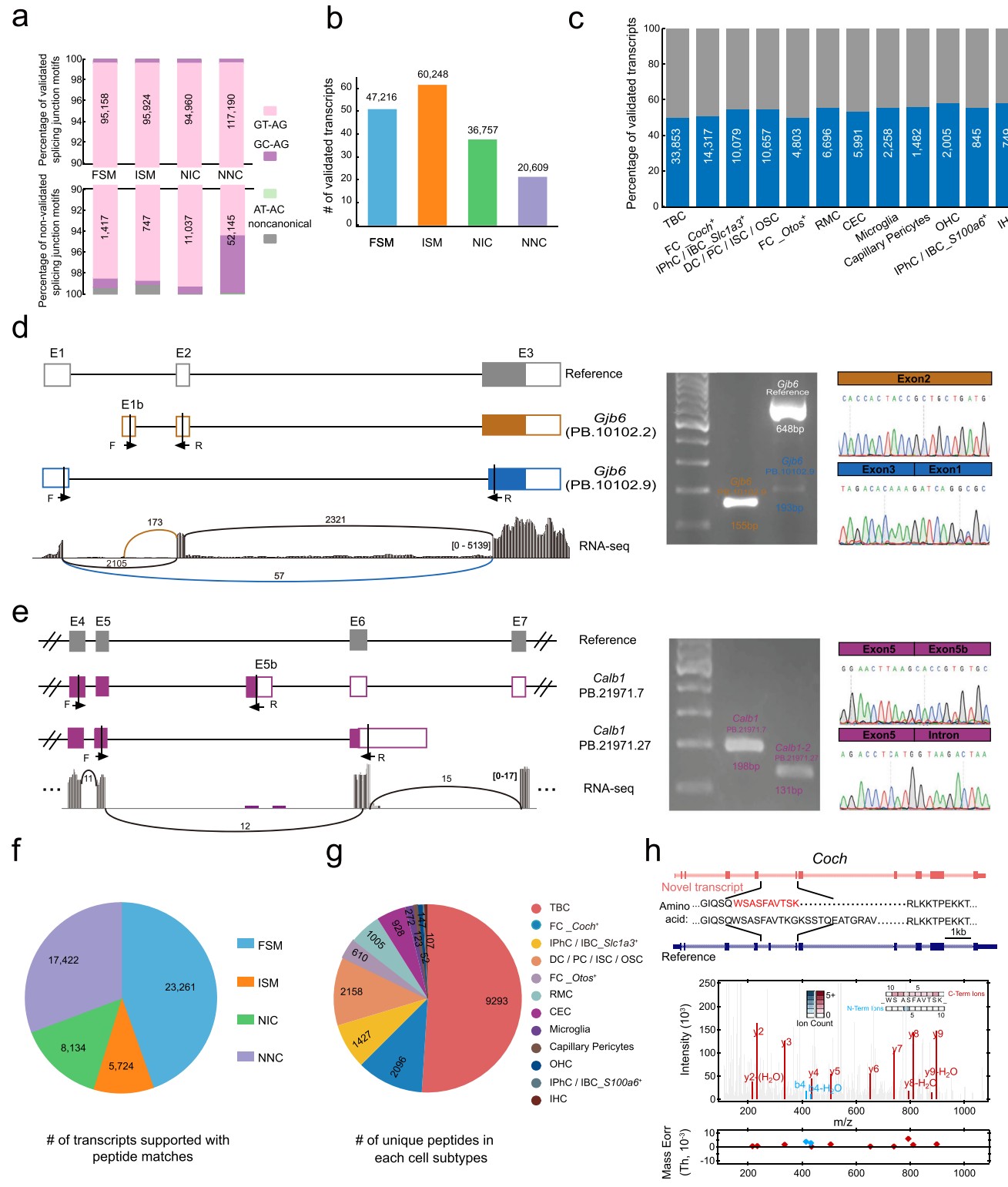

were generated: 1) Exon 2 knockout (KO) mice with disrupted canonical isoform expression (*Otof-ΔC*); and, 2) exon 6b KO mice targeting the expression of the short isoform (*Otof-ΔS*). *Otof⁻/⁻* mice (exon 14-15 deletions) served as a negative control, as neither the short nor the annotated isoform was expressed in this line (Fig. 4a–d). Expression of the different otoferlin isoforms was examined by performing whole-mount immunostaining using two otoferlin antibodies (recognizing the protein fragment encoded by exons 1–13 or exons 7-8, respectively; Fig. 4a–d). The canonical isoform was stained with both antibodies (Fig. 4a, c). The coding sequence of the short isoform overlaps with the

canonical one; thus, it cannot be specifically detected in wild-type (WT) IHCs. However, when examining exon-2 knockouts with the disrupted canonical isoform, IHCs from *Otof-ΔC* mice were successfully stained with only one antibody (recognizing the protein fragment encoded by exons 7-8). Moreover, no otoferlin staining was observed in the IHCs of the negative control mice (*Otof⁻/⁻*). In addition, the quantity of isoform expression was measured using qPCR and immunofluorescence staining of *Otof-ΔC* and WT mice. We found that the RNA expression level of this short isoform was comparable to that of the canonical one and fluorescence intensity of otoferlin was significantly reduced after

**Fig. 3 | Validation of the AS events of long-read sequencing data at both transcription and translation levels. a** Validation of splice junctions by bulk RNA-seq data. The proportions of validated and nonvalidated splicing junction motifs by RNA-seq in each category of transcripts (the number refers to the validated and nonvalidated splice motifs number). **b** The bar chart indicates the number of validated transcripts (all the junctions were validated) by RNA-seq. **c** The proportion of AS patterns varied in different cochlea cell subtypes. **d, e** Validation of the novel splicing isoforms of deafness-related genes using RNA-seq, reverse transcription-polymerase chain reaction (RT–PCR) and Sanger sequencing (representative results of at least three biological replicates are shown). Transcript structure and name are shown in the left panel, with protein-coding exons, noncoding exons, and introns represented by filled boxes, open boxes, and lines. The forward and reverse PCR primers are marked with lines and arrowheads in the relative transcript

structure. Sashimi plot showing the sequencing read coverage for differentially spliced events and novel splice junctions are color-coded. The gel bands in each figure show standard size markers and the expected size of PCR products, and each product identity was confirmed by Sanger sequencing. **f** The pie chart indicates the number of transcripts supported with peptide evidence. **g** The unique peptide expression of each cell type after mapping the total peptides to the cell cluster ScISOr-Seq data. **h** Examples of the unannotated isoform-specific peptides validated by mass spectrometry-based proteomics of *Coch*. The upper panel represents the novel transcript, amino acid sequences (the peptide indicated in red has been detected by proteomics), and the reference transcript. The fragment ions supporting the corresponding peptide are highlighted (red, y ion; blue, b ion) in the lower panel.

deleting the canonical isoform (Supplementary Fig. 4c, d). Together, these results demonstrate that the short isoform of otoferlin previously had been overlooked.

To investigate whether the Otof short isoform could provide significant new insights into auditory function, we first assessed ABR and found no noticeable threshold elevation in 1-month-old [$F_{(6, 196)} = 0.19$, $p = 0.98$] and 3-month-old [$F_{(6, 140)} = 0.71$, $p = 0.64$] WT and *Otof-ΔC* mice at all sound frequencies tested (Fig. 4d, e). This was distinctly different from previous studies in other otoferlin mutation models that exhibited profound hearing loss caused by depletion of overall otoferlin levels[24,25,47]. In contrast, *Otof$^{-/-}$* mice showed no visible ABR waves, consistent with the findings of a previous study[25] (Fig. 4e, f). The ABR threshold tests the functional integrity of OHCs that support normal OHC function of *Otof-ΔC* mice. This was further validated by distortion product otoacoustic emission (DPOAE) data (Supplementary Fig. 5a).

Successful synaptic transmission can be determined by normal ABR wave I amplitude and latency, which reflects the synchronized firing capability of auditory nerve fibers and the elapsed synaptic transmission time[49]. Of note, *Otof-ΔC* mice exhibited a significant reduction [$F_{(11, 336)} = 3.848$, $p < 0.001$] in the mean peak amplitude of the ABR wave I as compared with WT mice (Fig. 4g, Supplementary Fig. 5b, c). This auditory phenotype of *Otof-ΔC* mice was consistent with HHL, defined as an auditory neuropathy characterized by reduced auditory nerve input but normal hearing thresholds[50]. However, the discrepancy did not result from changes in the number of functional synapses (Supplementary Fig. 6) or synaptic delay as measured by ABR wave I latency [$F_{(11, 336)} = 0.2147$, $p = 0.99$] (Fig. 4h, Supplementary Fig. 5d).

To explore whether the short isoform knockout could induce hearing impairment, a *Otof-ΔS* mice was generated. No obvious ABR threshold elevation was found in *Otof-ΔS* mice [$F_{(6, 49)} = 0.1784$, $p = 0.99$] (Fig. 4e, f). Additional measurements of amplitude and latency of ABR wave I showed no changes in *Otof-ΔS* mice compared to WT mice (Supplementary Fig. 5e, f). In addition, the two heterozygous mutant mice lines also showed normal auditory brainstem responses (Supplementary Fig. 5g–l). The expression of otoferlin was also significantly decreased in *Otof-ΔS* and *Otof$^{+/-}$* mice (Supplementary Fig. 4c, d) but without hearing impairment. Thus, we concluded that the auditory phenotype of *Otof-ΔC* was not a mere expression of decreased otoferlin level, but was the null result of the canonical isoform.

To test whether presynaptic function accounted for the reduction in ABR wave I in *Otof-ΔC* mice, using a whole-cell patch-clamp configuration, we measured the amplitude of $Ca^{2+}$ currents ($I_{Ca}$) and the kinetics of $I_{Ca}$ activation and inactivation. No significant changes in WT and *Otof-ΔC* mice were detected, but a significant reduction in the $I_{Ca}$ and inactivation in *Otof$^{-/-}$* mice was seen (Fig. 4i, Supplementary Fig. 7a, b). To assess whether IHC exocytosis was impaired in *Otof-ΔC* mice, we recorded $Ca^{2+}$-triggered exocytosis in response to a depolarizing stimulus at various time-durations (Fig. 4j–l). The

evoked $ΔC_m$ of IHCs in response to short stimuli (<10 ms) was comparable between WT and *Otof-ΔC* mice. However, for longer stimulus durations, exocytosis was significantly attenuated in *Otof-ΔC* IHCs (Fig. 4k). In addition, the efficiency of $Ca^{2+}$-triggered exocytosis, as assessed by the ratio of $ΔC_m/Q_{Ca}$, was less efficient in *Otof-ΔC* IHCs (Fig. 4l). This discrepancy was independent of stimulus amplitudes (Supplementary Fig. 7c, d). As expected, all exocytosis was completely abolished in IHCs of *Otof$^{-/-}$* mice (Fig. 4j–l). Electrophysiological characterization of IHCs in *Otof-ΔS* mice, such as $Ca^{2+}$ current and exocytosis, revealed no obvious differences compared to WT mice (Supplementary Fig. 7e, f). Therefore, the *Otof-ΔC* mouse model, expressing only the short isoform, was able to support normal hearing thresholds but with reduced sustained exocytosis. This allowed to study the roles of otoferlin in synaptic vesicle release dynamics in IHCs, and to explore the possible reasons why the short isoform can not support normal auditory function, demonstrated in the following experiments.

## Unveiled previously unidentified function of otoferlin in endocytic membrane retrieval in IHCs only expressing the short *Otof* transcript

Sustained exocytosis not only depletes the readily releasable pool but also results in the replenishment of cytosolic vesicles at the release site. Impaired vesicle replenishment was observed under both paired-pulse (Fig. 5a) and pulse train stimuli (Fig. 5b). Otoferlin has been proposed to be involved in membrane retrieval. The kinetics of endocytosis have not been evaluated due to the lack of measurable exocytosis responses. The *Otof-ΔC* model, which only expressed the short transcript allows a more refined investigation of endocytosis, helping us address the question of whether otoferlin participates in the endocytic process. In a typical electrophysiological recording, a protracted linear component after a depolarizing stimulus likely reflects clathrin-mediated endocytosis[51]. In *Otof-ΔC* mice, we found a significantly shallower slope (Fig. 5c). These data proved that otoferlin was directly involved in vesicle endocytosis in IHCs. This is in contrast to previous notions that endocytosis was normal in the IHCs of *Otof$^{C2C/C2C}$* and *Otof$^{Pga/Pga}$* mice[24,48].

We further characterized deficits in *Otof-ΔC* mice by ultrastructural analysis of presynaptic vesicles. The number of cytosolic vesicles was found to decrease with or without depolarizing high-potassium incubation, and endosome-like vacuoles (ELVs) were found in the IHCs of *Otof-ΔC* mice (Supplementary Fig. 8), indicating impaired endocytic processes[52,53]. In summary, otoferlin isoforms can help characterize the biological functions of *Otof* in normal presynaptic vesicle recycling.

To verify our findings in vivo, we designed an ABR experiment aimed at recapitulating cellular events. In a normal ABR recording, ABR responses were averaged 400 times to achieve a good signal-to-noise ratio. The stimulus repetition rates were typically set to ~20/s to avoid fatigue. In this experiment, we increased the stimulus rate to 60/s (maximum allowed in the ABR recording setting without truncating

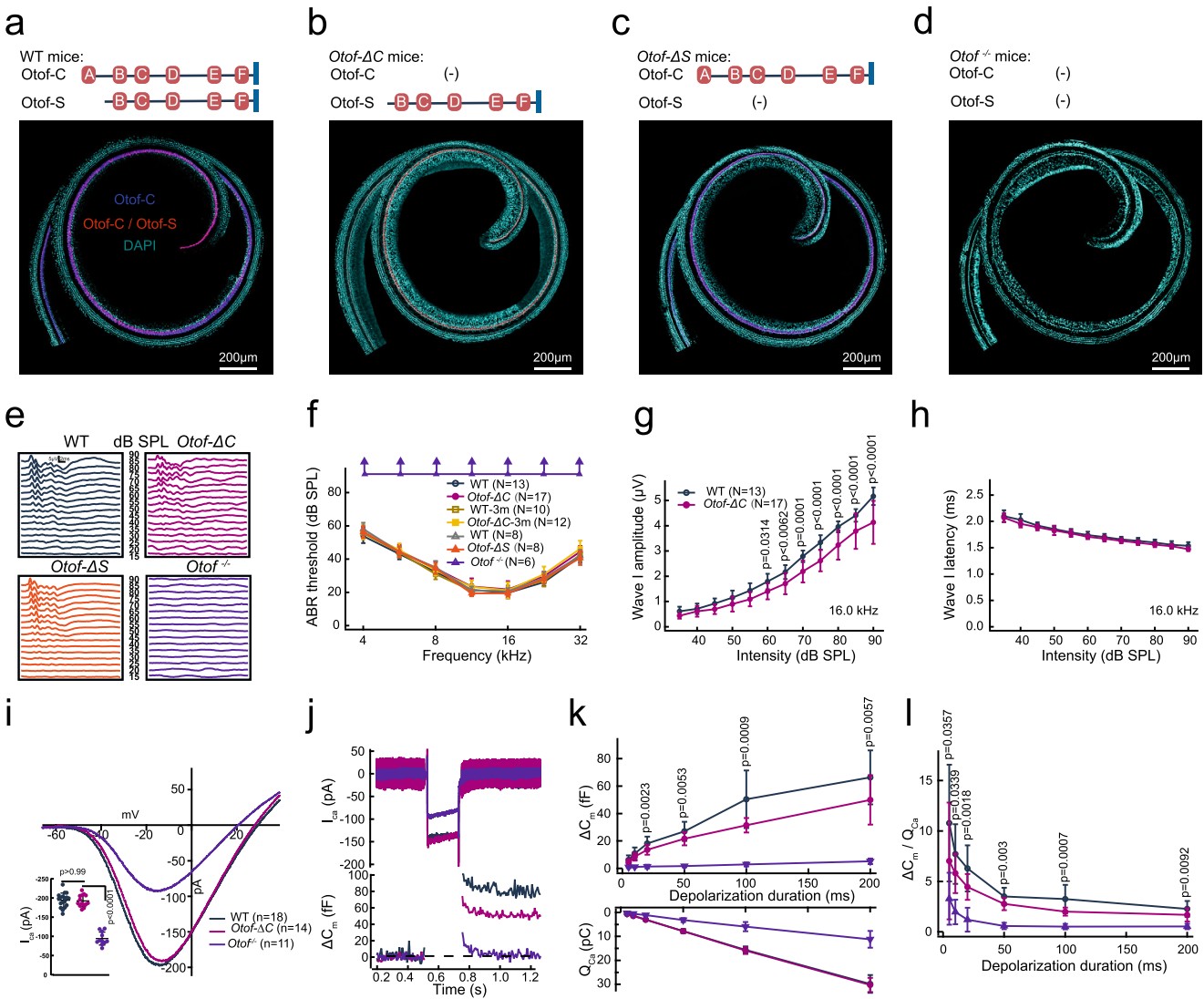

**Fig. 4 | A short isoform of otoferlin is expressed in the cochlea, and mice expressing only the short transcript show a normal hearing threshold but impaired auditory nerve fiber responses due to decreased exocytosis of IHCs.** **a–d** Different otoferlin isoforms are illustrated in the organs of Corti from WT, *Otof-ΔC, Otof-ΔS,* and *Otof* [-/-] mice at P28 (representative results of at least three biological replicates are shown). IHCs were double-stained with two antibodies (one targeting the C₂A- B regions (exons 1-13) and another targeting the region before the C₂B domain (or exons 7-8), staining Otof-C and Otof-C + Otof-S respectively) in WT and *Otof-ΔS* IHCs, single-stained (Otof-S) IHCs were found in *Otof-ΔC* mice, and a complete loss of staining was observed in *Otof* [-/-] mice. **e** Example of ABR waveform series at different sound intensities (16.0 kHz, 90–15 dB SPL) in WT, *Otof-ΔC,* and *Otof* [-/-] mice. **f** The ABR thresholds for both WT and *Otof-ΔC* mice were normal (1-month-old and 3-month-old), while the *Otof* [-/-] mice exhibited profound hearing loss (arrowhead: showed no response at 90 dB SPL). **g, h** ABR wave I amplitude and latency were plotted as a function of sound levels (16.0 kHz). *Otof-ΔC* mice revealed

a significant reduction in ABR wave I amplitude but normal latency. **i** The $Ca^{2+}$ current-voltage curves showed comparable $Ca^{2+}$ currents in *Otof-ΔC* and WT IHCs but significantly reduced $Ca^{2+}$ currents in *Otof* [-/-] mice. **j** Representative $Ca^{2+}$ currents (top) and corresponding membrane capacitance ($\Delta C_m$) traces were recorded from IHCs from the three genotypes. **k** $\Delta C_m$ (upper) and calcium influx ($Q_{ca}$, lower) evoked by depolarizing pulses with durations ranging from 5 to 200 ms. $\Delta C_m$ for prolonged stimulation was reduced in IHCs from *Otof-ΔC* mice compared with WT mice. **l** The efficiency of $Ca^{2+}$ in triggering exocytosis, assessed based on the ratio of $\Delta C_m/Q_{Ca}$, was reduced significantly in *Otof-ΔC* IHCs compared with WT IHCs. Age-matched and sex-matched littermate WT controls were used for all experiments. Statistical analysis by two-way ANOVA followed by the Bonferroni *post hoc* test with significance indicated (**g**), one-way ANOVA followed by the Bonferroni *post hoc* test with significance indicated (**i**) and two-side unpaired *t* test or Mann–Whitney test with significance indicated (**k, l**). All data, statistical test used and *p* values can be found in the source data file.

recording traces), similar to the pulse train stimulus applied to IHCs. At the selected 16 kHz 90 dB SPL stimulus, a robust ABR response with clear separation of all initial four peaks was easily achieved with a lower average number. To stress the vesicle release and recycling process, ABR wave I changes were tracked by increasing the average numbers (Fig. 5d). Comparison of wave I amplitudes by incrementing average numbers from 50 to 12,800 showed a significant rundown of the wave I amplitude in *Otof-ΔC* mice but no decline in the WT mice (Fig. 5e). Taken together, these studies of alternative isoforms in the cochlea indicate a seminal role otoferlin plays in endocytosis function, which

has not previously been well-characterized. Furthermore, we have uncovered the mechanism of HHL due to impaired vesicle recycling of IHCs.

## The otoferlin short isoform showed decreased binding ability to synaptic proteins

Membrane retrieval following exocytosis in a timely and highly efficient manner is essential for the indefatigable synaptic transmission of IHCs. It involves both the interaction of "exocytic" and "endocytic" proteins[54]. Otoferlin, a central exocytic protein at IHC active zones, is

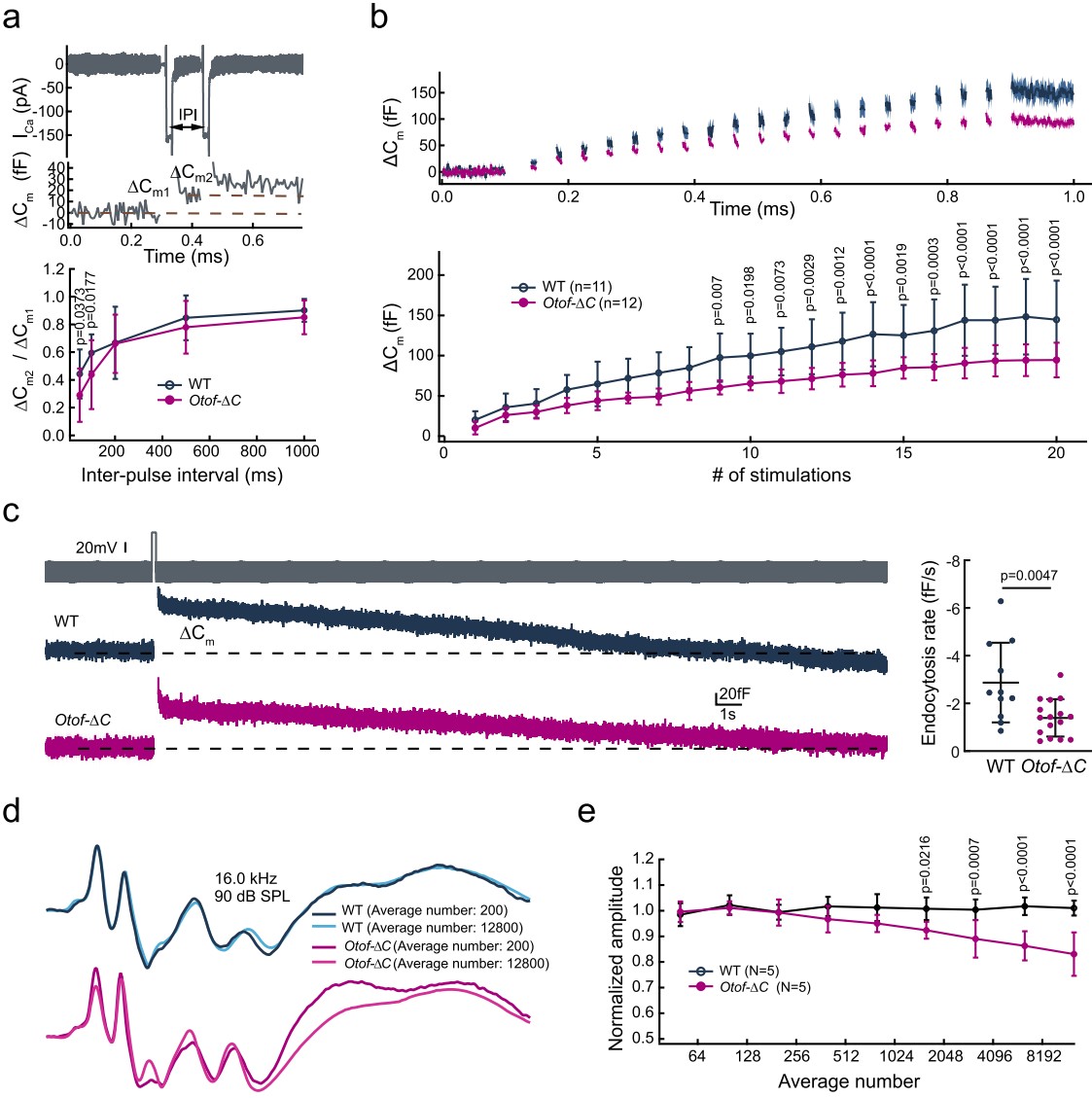

**Fig. 5 | Reduced vesicle replenishment and slowed endocytic membrane retrieval in *Otof-ΔC* IHCs. a** Representative trace collected from the double-pulse stimulation revealed a notable $\Delta C_m$ after two consecutive 20 ms depolarizations. The ratio of $\Delta C_{m2}/\Delta C_{m1}$ was calculated to quantify synaptic vesicle replenishment. Significantly reduced $\Delta C_{m2}/\Delta C_{m1}$ was found in *Otof-ΔC* IHCs at 50 and 100 ms only (lower). **b** A train of twenty successive short depolarization pulse (duration 5 ms, interpulse interval 10 ms) induced $C_m$ increments (upper panel) with IHCs from *Otof-ΔC* mice showed a cumulative decline in $\Delta C_m$. **c** Endocytosis was assessed by determining the decrease in $C_m$ after 200 ms depolarizing pulses and fitting with a linear function. *Otof-ΔC* IHCs showed a slower linear Cm recovery after a 200 ms depolarization. **d** Comparison of ABR wave I amplitudes with different numbers of

averaging repeats showed no change in WT but a reduction in wave I amplitude in *Otof-ΔC* mice, suggesting a slower recovery in triggering auditory nerve fiber firing. **e** Increments from 50 to 12800 ABR averages were applied to a fixed stimulus condition of 16 kHz, 90 dB SPL. As the number of averages increased, a cumulative reduction in the ABR wave I amplitude recapitulated the train pulse stimulus results from IHC patch-clamp recordings. Age-matched and sex-matched littermate WT controls were used for all experiments. Statistical analysis by two-side unpaired *t* test or Mann–Whitney test with significance indicated (**a**, **c**) and two-way ANOVA followed by the Bonferroni *post hoc* test with significance indicated (**b**, **e**). All data, statistical test used and *p* values can be found in the source data file.

an obvious candidate for regulating exocytosis-endocytosis coupling. The short isoform encodes an N-terminal truncate protein lacking the C2A domain, and therefore, it was inferred that the short and canonical isoforms might have different protein binding abilities[55].

As suggested above, we investigated whether the otoferlin short isoform could interact with endophilin A1, which promotes vesicle recycling[56]. We employed a pull-down assay using lysates prepared from HEK293T cells expressing Otof-C-HA or Otof-S-HA with purified Gst-Endophilin A1. Notably, as seen with the canonical isoform[56], the short isoform can also interact with endophilin A1 (Fig. 6a). Taking into consideration the tissue-specificity expression of otoferlin, we performed immunofluorescence staining of IHCs. We found that endophilin A1 can colocalize with otoferlin in WT and *Otof-ΔC* mice,

indicating they form an in vivo complex (Fig. 6b). Otoferlin is a multi-C2 domain (C2A-F) protein. The C2A domain was unable to bind Ca²⁺ but does interact with other proteins[55]. Thus, we hypothesized that the short isoform interaction with endophilin A1 might be weakened. To obtain an estimate of the extent to which binding is affected, we employed GST pull-down assays performed with recombinant Gst-endophilin A1 and His6-otoferlin fragments (C2ABC, C2BC, and C2A). The results showed that C2ABC, C2BC and C2A could directly bind to endophilin A1. Compared to C2ABC, the amount of C2BC co-immunoprecipitated with endophilin A1 was reduced (Fig. 6c). C2B-F has the ability to bind[57] to Ca²⁺. Next, we investigated the effects of calcium on binding. Under different Ca²⁺ concentrations, no significant change was found in binding affinity between otoferlin (or its

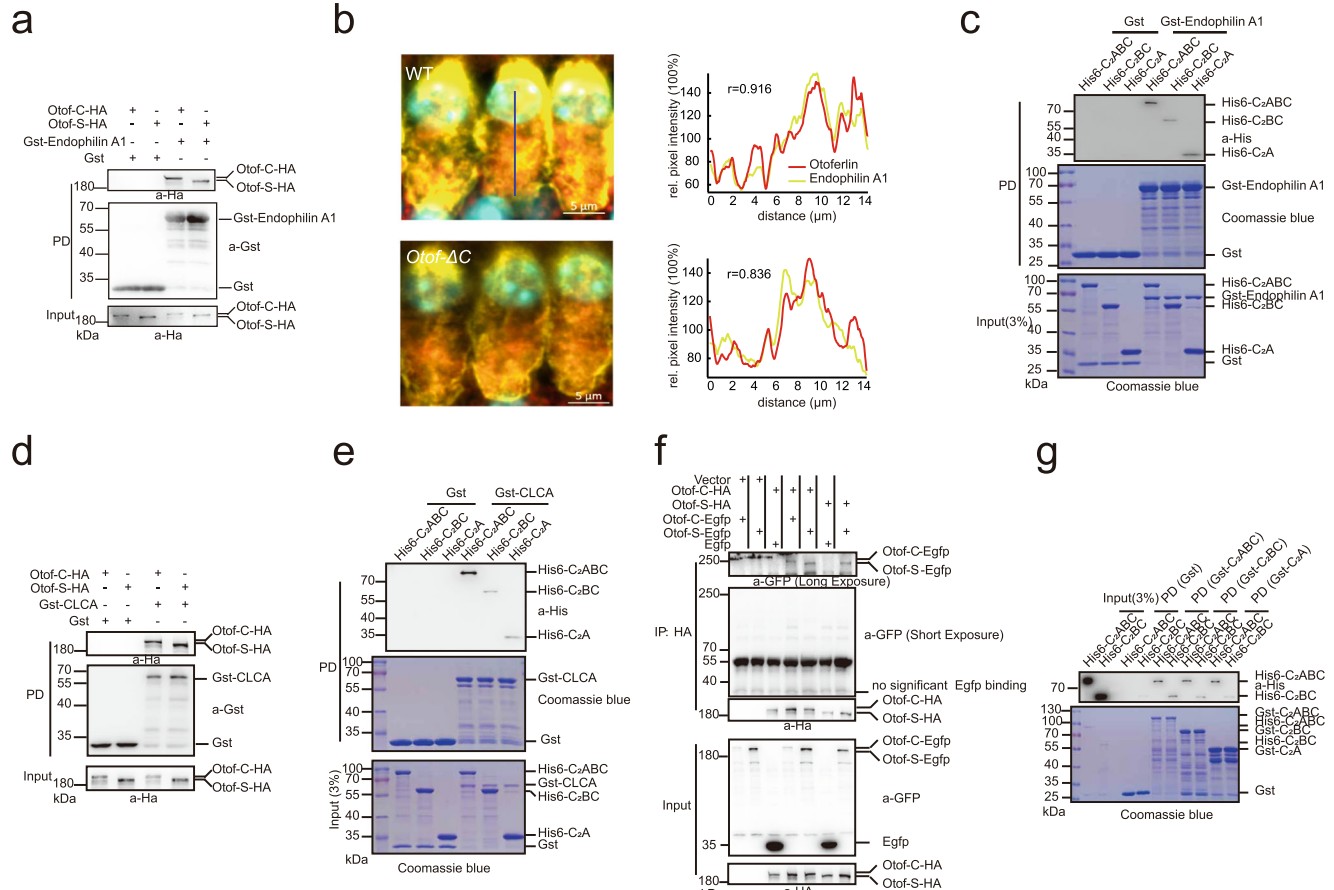

**Fig. 6 | Protein interaction ability is decreased of the otoferlin short isoform.**
**a** HEK293T cells were transfected with constructs expressing HA-tagged Otof-C/S for 48 h and lysed for pull-down assay with purified Gst-Endophilin A1. Input and pull-down proteins were analyzed by immunoblotting with antibodies against HA and GST, and showed that Otof-C/S could interact with endophilin A1. **b** Endophilin A1 (yellow) and otoferlin protein (red) expression in IHCs (left). Normalized intensity (normalized to background immunofluorescence) profiles are drawn from the blue lines (left) and show the relative pixel intensity along the line (right). Colocalization was analyzed by Pearson correlation of the two channels. **c** GST-tagged endophilin A1 conjugated to glutathione-Sepharose was incubated with purified His6-tagged $C_2ABC$, $C_2BC$, and $C_2A$ proteins. Input and pull-down proteins were analyzed by coomassie blue-stained SDS-PAGE gel and immunoblotting

antibodies against His. The binding ability with endophilin A1 was reduced in $C_2BC$ protein. **d** Pull-down assay of HA-tagged Otof-C/S with purified Gst-CLCA. The result showed that Otof-C/S could interact with clathrin light chain A. **e** GST-tagged CLCA conjugated to glutathione-Sepharose was incubated with purified His6-tagged $C_2ABC$, $C_2BC$, and $C_2A$ proteins, and the interaction with CLCA was decreased in $C_2BC$ protein. **f** The interaction of Egfp-tagged Otof-C/S or HA-tagged Otof-C/S was detected by a coimmunoprecipitation assay using an anti-HA antibody. The result showed that otoferlin could bind with itself. **g** Binding of His-tagged $C_2ABC$, $C_2BC$ to GST control and GST-$C_2ABC$, GST-$C_2BC$, and GST-$C_2A$ in the GST pull-down assay. The self-interaction was also notably reduced. Assays were performed at least three times.

fragments) and endophilin A1 (Supplementary Fig. 9a–d). Decreased binding between $C_2BC$ and endophilin A1 was independent of $Ca^{2+}$ concentrations manner. Collectively, these results suggest that the interaction between Otof-S and endophilin A1 was not completely abolished but strongly reduced, likely due to the N-terminal amino acid truncation, which implies an auditory functional impairment of *Otof-ΔC* mice.

Clathrin, which plays a key role in clathrin-mediated endocytosis[58,59], is another example of the attenuated protein binding ability of the otoferlin short isoform. We showed that otoferlin could directly bind to clathrin light chain A (CLCA, Fig. 6d, Supplementary Fig. 9e). Similar to the results obtained with endophilin A1, the $C_2BC$ showed diminished binding to CLCA (Fig. 6e). In addition, C2 domains also interacted with each other[55,60,61], which may be functionally relevant. As expected, otoferlin interacted with itself (Fig. 6f). The interactive ability between $C_2BC$ and $C_2BC$ was much lower than that of $C_2ABC$ and $C_2ABC$ (Fig. 6g). This suggests that otoferlin fragments containing the $C_2A$ domain can enhance protein binding ability (Fig. 6g, Supplementary Fig. 9f). However, whether and how self-interaction impacts otoferlin function remains in question. Overall,

these observations, together with electrophysiology findings in *Otof-ΔC* mice, help us understand the differences between the two isoforms and may explain why the otoferlin short isoform cannot support normal membrane retrieval.

## The ratio of the different otoferlin isoforms underlying IHC functional differences in auditory processing

In neurons, dynamic changes in AS have been shown to either support neuronal functional plasticity, or result in disease pathology[62]. To illustrate whether, and to what extent, these isoforms participate in biological and pathological functions, we measured the otoferlin isoform ratio (Otof-C/(Otof-C+Otof-S)) by comparing immunofluorescence intensity. The hearing organ is tonotopically organized such that characteristic sound-wave frequencies in hair cells gradually change with cochlear position. We found that the otoferlin isoform ratio exhibited tonotopic differences, and from the apex toward the base of the cochlea, expression of Otof-C was notably increased relative to that of Otof-S (Fig. 7a–c), manifesting the functional difference of IHCs. To verify this, whole-cell patch-clamp recordings were performed. We observed that sustained exocytosis of synaptic vesicles in

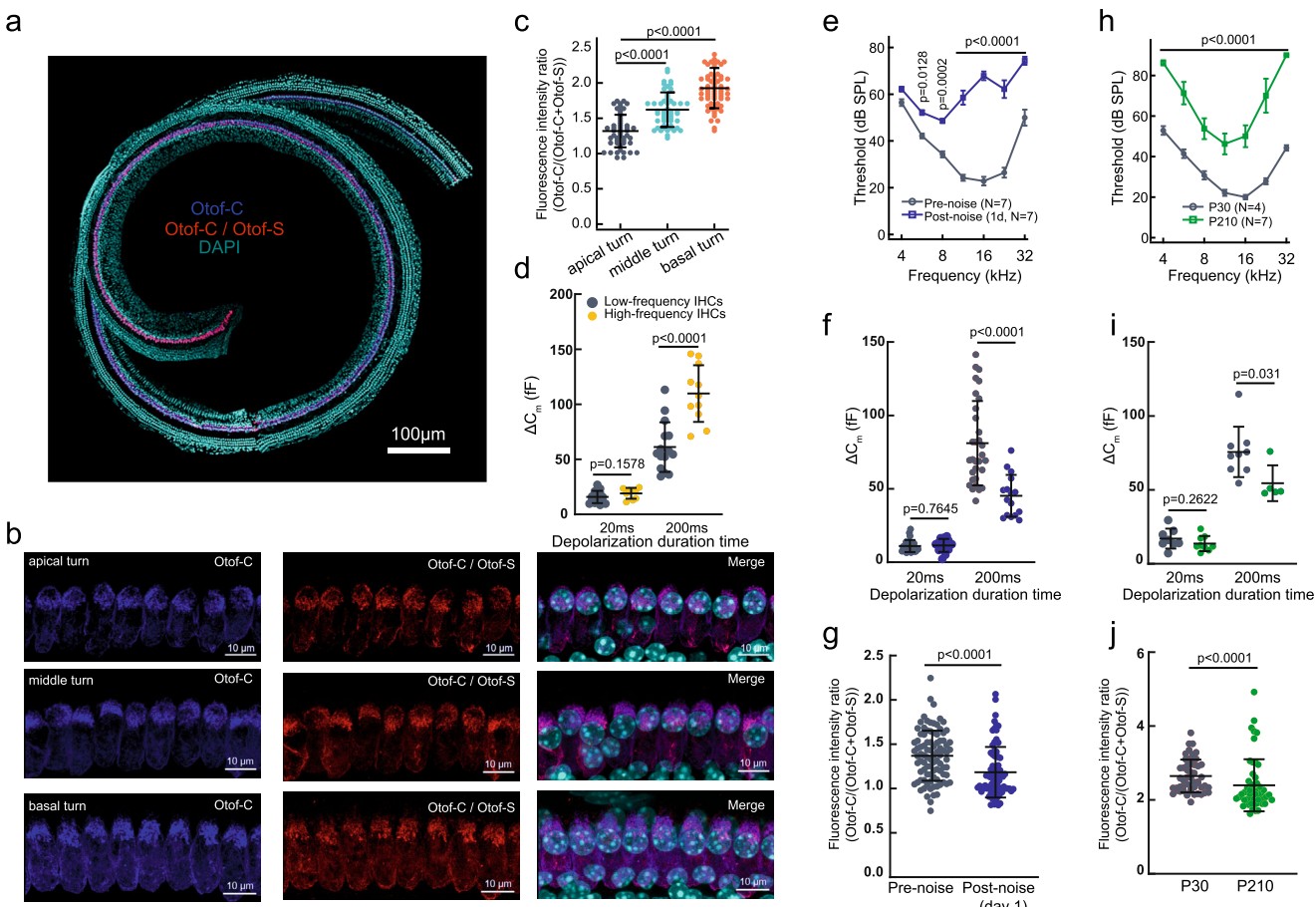

**Fig. 7 | The proportion of otoferlin isoforms was altered under noise exposure and aging conditions. a, b** Maximum intensity z-projections of confocal sections of IHCs with double labeling for otoferlin isoforms from P30 wild-type whole-mount organ of Corti preparations (representative results of at least three biological replicates are shown). **c** Otof-C immunofluorescence density ratio (intensities were normalized against a dark background in the image and averaged per cell) in IHCs shows a significant discrepancy between the apical (about the 8 kHz region, $n = 49$), middle (about the 16 kHz region, $n = 51$), and basal turn (about the 32 kHz region, $n = 66$) from 8 mice. **d** The sustained exocytosis of IHCs was significantly higher in high-frequency regions than in low-frequency regions. **e–g** An episode of 2 h, 103 dB SPL, and 2–20 kHz bandpass noise exposure significantly elevated ABR thresholds within 24 h. The sustained exocytosis of synaptic vesicles was

significantly reduced one day after noise exposure (**f**). The normalized Otof-C immunofluorescence density ratio in IHCs shows a significant discrepancy between pre-and post-noise exposure (a total of 92 and 77 cells of 8 mice from each group) (**g**). **h–j** Mice at 7 months old (P210) showed a significant elevation in ABR thresholds. The sustained exocytosis of synaptic vesicles was significantly reduced in aging mice (**i**). The normalized Otof-C immunofluorescence density ratio in IHCs showed a significant discrepancy between young and old mice (a total of 51 and 52 cells in 6 mice from each group) (**j**). Statistical analysis by one-way ANOVA followed by the Bonferroni *post hoc* test with significance indicated (**c**), two-way ANOVA followed by the Bonferroni *post hoc* test with significance indicated (**e, h**) and two-side unpaired *t* test or Mann-Whitney test with significance indicated (**d, f, g, I, j**). All data, statistical test used and *p* values can be found in the source data file.

IHCs was greatly increased toward the high-frequency region of the cochlea (Fig. 7d). Therefore, it was inferred that the expression of spliced isoforms along the tonotopic axis might enable IHCs to maintain sustained transmitter release under high-frequency stimulation.

The impact of environmental factors on the isoform ratio was examined by choosing a 2 h, 103 dB SPL 2–20 kHz bandpass noise to stress the mouse cochlea. The ABR thresholds were significantly elevated one day after noise exposure (Fig. 7e), and exocytosis of IHCs was significantly reduced (Fig. 7f). Measurements of the immunofluorescence intensity ratio of Otof-C, which reflect the changes in the proportion and ratio, were significantly changed after noise exposure (Fig. 7g). Therefore, it was also inferred that expression reduction of Otof-C may correlate with a reduction of vesicle release (Fig. 7f). Aging is another factor contributing to hearing loss and cochlear synaptopathy[63]. We found a significant elevation in the hearing threshold and reduction in synaptic exocytosis in 7-month-old mice (Fig. 7h, i). An alternation of the Otof-C immunofluorescence intensity ratio was also observed in the aging

cochlea, a trend similar to that seen in noise exposure (Fig. 7j). In summary, these results provided new insights into the mechanisms of auditory physiological and pathophysiological processes at the isoform level.

## Discussion

These findings add to an already compelling literature highlighting the inherent complexity of cochlear gene transcription and auditory function. Building on well-characterized cell types and cellular heterogeneity at the transcription level in the cochlea[26,64,65], we focused on mRNA transcripts to study consequences in the auditory periphery. To our knowledge, the present study represents comprehensive characterization of the transcripts expressed in the inner ear cochlea using large-scale long-read RNA sequencing across different cellular subtypes. We have found abundant transcript isoform diversity, and enriched cochlea genome annotation, which provides a powerful auditory research tool. These experiments confirm the importance of identifying alternative isoforms in the cochlea and highlight its important mechanistic role in auditory processing.

Genome annotations were usually constructed de novo from short-read RNA-seq, with or without guidance of existing genome assembly. While RNA-seq-based annotations are generally sufficient to identify protein-coding mRNAs and characterize their expression levels in large RNA samples, they are limited in their ability to fully annotate complete transcript isoforms, extended UTRs, and lncRNAs. With the ability to sequence long RNA molecules in a single read, Iso-Seq overcomes these limitations and has been used to identify unannotated transcript isoforms in a variety of genomes, including the well-annotated human genome[66]. Previous work by Ranum et al.[67] was the first to explore the inner ear transcriptome by using a low-throughput approach combining short- (Smart-seq2) and long-read (Nanopore) sequencing with handpicking strategies. While they only focused on the full-length transcriptome of OHCs in the inner ear, several cell-types such as IHCs and supporting cells were missing in their work. In our study, we integrated short-read (Illumina) and long-read (PacBio) sequencing high-throughput method, and the transcriptomic landscape of 12 cell-types in the inner ear were explored. However, the gene numbers per cell were lower than the previous work[67] due to insufficient data volume, and the methodology still needs to be improved.

The results of the current study provide a valuable resource for gene annotation in the inner ear. Compared to the GENCODE annotation (average of 2.56 isoforms per gene), the inner ear cochlea showed a surprisingly diverse isoform level with an average of 9.75 novel isoforms per gene. A total of 53.65% of the detected transcripts were novel, i.e., not listed in the GENCODE reference (VM23), suggesting that existing annotations may be insufficient to characterize the cochlea at the transcriptional level. Transcript splicing underlies much of the diversity in expression in complex organisms. Several splicing regulators, such as $Rbm24$[68] and $Srrm4$[69], have been shown to be highly expressed in the hair cells to regulate mRNA stability and hair cell-specific alternative splicing. As a result, cells in the cochlear have unique compliments of transcripts, leading to specific protein expression in one cell type relative to another. This may have been partially due to functional differences, such as sensory cells performing mechanoelectrical transduction, and nonsensory cells maintaining ionic homeostasis[45]. Our research expanded our understanding of transcriptome diversity in the cochlear; however, we cannot exclude the presence of potential transcriptional noise, which awaits future studies to verify. Overall, these findings enrich the transcriptional information on the mouse genome elucidating auditory heterogeneity in functional studies.

Among the deafness genes, the first FDA-approved gene therapy product for clinical trials was treating otoferlin mutation-induced DFNB9, such as DB-OTO (https://hearinghealthmatters.org/hearingnewswatch/2022/decibel-therapeutics-clinical-trial-submission), OTOF-GT (https://www.biopharma-reporter.com/Article/2022/11/07/sensorion-receives-us-fda-rare-pediatric-disease-designation-for-gene-therapy) and AK-OTOF (https://akouos.com/). We have identified a functional otoferlin short isoform supporting normal hearing thresholds, which may have important implications for the usage of gene therapy instead of the long isoform. Although AAV-mediated gene transfer of the otoferlin short isoform is technically challenging due to the limited DNA packaging capacity (≈4.7 kb)[70], it may be useful in rescuing hearing loss using functionally truncated proteins. Identifying a functional isoform that can be transduced by AAVs may be an alternative way to treat some forms of deafness or other hereditary diseases.

To investigate whether the unannotated isoforms can provide insight into gene function, we focused on $Otof$, a well-known deafness-associated gene. This short transcript encodes a short protein isoform lacking the $C_2A$ domain, which serves as an excellent example of the value and importance of using transcript information to interpret otoferlin function. Previous, efforts to decipher the function of otoferlin involved several different animal models, such as $Otof^{-/-}$, $Otof^{Pga/Pga}$, $Otof^{Ala515,\ Ala517/Ala515,\ Ala517}$, and $Otof^{I515T/I515T}$ mice. In these animals, otoferlin was shown to function in synaptic vesicle exocytosis and replenishment[24,25,48,71]. However, little was known about the molecular regulation of hair cell endocytosis. In contradistinction to previous findings calming no effect on endocytic membrane retrieval[51,72], we observed that IHCs expressing only the $Otof$ short transcript exhibited deficits in synaptic vesicle endocytosis with normal calcium influx. These electrophysiological differences did not arise from cellular otoferlin content because $Otof^{Pga/Pga}$ IHCs also had a much lower otoferlin content but showed normal endocytic membrane retrieval[24]. We may need to consider the isoform-specific phenotypes to interpret this difference because we knock out the otoferlin canonical isoform. Although the $C_2A$ domain does not have $Ca^{2+}$-binding potential, it may interact with other proteins[60]. In this study, we found that the otoferlin short isoform could interact with endophilin A1, CLCA, and itself; however, binding ability was distinct from the known isoform, which may provide insight into understanding otoferlin function involving endocytic membrane retrieval.

Our findings showed that the short transcript supported a normal hearing threshold but with an impaired auditory nerve response. This is in contrast to previous notions of a significant threshold elevation due to the mutation affected both the two isoforms[24,25,48,71]. For example, the $Otof^{Pga/Pga}$ mice carry a N-ethyl-N-nitrosourea–mediated otoferlin missense mutation leading to an amino acid substitution (Asp to Gly) close to the C terminus of the protein;[73] thus, this mutation affects both the canonical and the short isoform. It is, therefore, possible that the loss of canonical isoform mainly causes some aspects of the described phenotype because the auditory function was normal in $Otof^{-\Delta S}$ and $Otof^{+/-}$ mice. The $Otof$-$\Delta C$ mice may represent a suitable animal model to address mechanism of hidden hearing loss (HHL). Up until now, the loss of synaptic connections and peripheral demyelinating disorders resulting from either noise exposure or aging have been shown to be mechanistic drivers of HHL[74,75]. The present study further demonstrates that presynaptic dysfunction can result in permanent auditory deficits characteristic of HHL. This cochlear synaptopathy is not associated with synaptic loss but rather with impairment in synaptic vesicle recycling in IHCs. This is the result of deletion of the otoferlin canonical isoform, which may have clinical implications in the diagnosis of HHL. Among the mutations of $Otof$, those classified as pathogenic or likely to be pathogenic have been shown to be mainly located downstream of exon 8[76]. Although the significance of the two otoferlin isoforms for human hearing impairments remains to be shown, we suggest that people with nonsense mutations located in exons 1–6 may suffer from HHL.

The existence of functional transcripts may be a compensatory mechanism for maintaining normal auditory functions. We found that the proportion of otoferlin isoforms varies tonotopically; Otof-C proportion was increased with the functional demands of detecting high-frequency sound. One explanation for this is that the otoferlin short alternative isoform has a lower exocytosis-endocytosis coupling efficiency than that of the canonical one. Moreover, our findings suggest potential molecular mechanisms involved in the sustained firing capability of IHCs in response to noise exposure and aging. The short isoform cannot fully support normal hearing functions, as shown in the $Otof$-$\Delta C$ mice, which had smaller wave I amplitudes. It is conceivable that under environmental stress, such as noise exposure conditions, the expression of functionally competent canonical isoforms becomes reduced, leading to a deficiency in neurotransmitter release. Hence, we observed that Otof-C proportions decreased as IHCs failed in meeting high frequency sound-wave stimulus demands. Similarly, these dynamic events were also observed in aging cochlea, providing new insights into auditory pathophysiology at the level of isoform resolution.

Current limitations in long-read sequencing are found in its moderate sequencing depth compared to short-read sequencing. This impedes the accuracy of identifying and quantifying the less abundant

mRNAs, such as *Myo15*, *Xirp2*, etc., and future investigations still needed to overcome this limitation. Thus, our Iso-seq experiments mainly focused on discovering uncharacterized transcript isoforms rather than their expression levels. We used the organ of Corti from postnatal Day 7 mice for long-read sequencing, as it was a significant bottleneck to performing single-cell sequencing and obtaining high-quality, full-length mRNA from adults[65]. Gene expression levels varied largely between early stages and young adults[33,77–79]. Yet, the category of alternative transcripts appeared to be constant during auditory development based on the validation experiments (bulk RNA-seq) performed using mice with peripheral auditory maturity. Overall, this study expands knowledge of splice isoform diversity and its possible functions. It affords a better understand the complex molecular mechanisms underlying auditory function in both normal physiological and pathological states.

## Methods

### Animals
Wild-type (C57BL/6 J) and genotypic mice of both sexes (obtained from Shanghai Model Organisms Center, Inc., Shanghai, China) were used for experiments. Mice were housed for the duration of these experiments in the animal care facility of the Ear Institute of Shanghai Ninth People's Hospital, in affiliation with the Shanghai Jiao Tong University School of Medicine. Mice were maintained in a dark/light cycle of 12 h / 12 h. Animals were kept at room temperature with a range of 22–25 °C with humidity of the animal room ranging between 50–60%. The experimental protocol was approved by the Institutional Animal Care and Use Committee at Shanghai Ninth People's Hospital (SH9H-2022-A926-1) and followed the guidelines for the Care and Use of Laboratory Animals (8th edition), published by the National Institutes of Health (Bethesda, MD, USA). Mice were euthanized by isoflurane (RWD, China) overdose via inhalation before tissue collection.

### 10x Genomics single-cell capture and single-cell library preparation
Single-cell capture and cDNA amplification of postnatal day (P)7 mice organ of Corti were performed using Chromium Single Cell 3' V3Reagent Kits (10× Genomics, USA) and Gel Beads with the following modifications. In brief, the RT time was increased to 2 h to potentially increase the efficacy of reverse transcription of longer transcripts. Finally, the amplified cDNAs were split into two pools, one pool was used for Illumina 3' sequencing for differential gene expression analysis, and the other pool was used for PacBio long-read sequencing to detect full-length RNA isoforms.

Illumina library preparation was performed using 100 ng of amplified cDNA following the Chromium Single Cell 3' Reagent Kits V3 User Guide. The final libraries were sequenced on the Illumina NovaSeq 6000 platform (Illumina, USA). Pacbio library preparation was performed using the SMRTbell™ Template Prep Kit 2.0 (Pacific Biosciences, USA), and then assessed using an Agilent Bioanalyzer 2100 system (Agilent Technologies, USA) and Qubit HS (Life Technologies, USA) before sequencing. SMARTbell sequencing libraries were bound to polymerases using the Sequel Binding Kit 2.1 and V3 Primers. The polymerase template complexes were bound to MagBeads with the PacBio MagBio Binding Kit. Sequencing reactions were performed using the PacBio Sequel II platform (Pacific Biosciences, USA) in CCS mode.

### Illumina short-read data analysis
Illumina short reads were aligned and quantified using the Cell Ranger Single-Cell Software Suite (v.3.0.1, 10x Genomics) and the mouse reference genome (mm10). The output gene count matrix from the Cell Ranger pipeline was analyzed in RStudio (v 1.3.1056) with R (v 4.2.2) using the packages Seurat (v4.3.0; https://github.com/satijalab/seurat) for the standard Seurat pipeline. In brief, the 'SCTransform'

algorithm was used to normalize and scale the data[80]. PCA was performed and unsupervised clustering was applied to the top 50 PCs and the clustering resolution was 0.2. Cell type specific markers were identified using Seurat's "FindALLMarkers" with default settings.

### ScISOr-Seq data analysis
**Generation of circular consensus sequencing reads.** The PacBio raw reads were analyzed using the SMRT Link (v 8.0) pipeline. High-quality circular consensus sequence (CCS) reads were generated with the following modified parameters: "--min-passes 0 --min-length 50 --max-length 21000 --min-rq 0.75".

**Generation of single cell full-length non-concatemer (FLNC) reads.** For each read, we located the barcode sequence by searching for the flanking sequence before the cell barcode. The cell barcodes identified from the short-read data provided a reference to search for and trim in these long reads. Reads that failed to match any cell barcode were discarded.

Reads containing the adaptor primers as well as a poly-(A) tail at the 3' end were considered as FLNC reads. Lima (v1.10.0) and lsoseq3-refine (v3.2.2) were used on ccs.bam file separately for isolating FLNC reads using the following parameters: lima ccs.bam \ PB_adapters.fasta \ fl.bam \--isoseq \--num-threads 12 \ --min-score 0 \ --min-end-score 0 \ --min-signal-increase 10 \ --min-score-lead 0, isoseq3 refine fl.primer_5p--primer_3p.bam \ PB_adapters.fasta \ flnc.bam \ --min-polya-length 20 \ --require-polya.

FLNCs were aligned to the mouse genome reference (mm10) using minimap2 (v 2.2)[81] with the parameters "-ax splice -uf --secondary=no -C5 -O6,24 -B4". The "collapse_isoforms_by_sam.py" script in "cDNA_Cupcake (v 8.2)" was used to filter out redundant and low confidence reads with low mapping quality. The PB number represented a specific isoform per gene locus.

**Non-redundant isoforms classification.** SQANTI3 (v 4.0) (https://github.com/conesalab/SQANTI3) was used to classify non-redundant transcripts into four categories: FSM (full splice match), ISM (incomplete splice match), NIC (novel in catalog) and NNC (novel not in catalog). To identify open reading frames in the transcript sequences, GeneMarkS-T[82] (v 5.1) was implemented in SQANTI. We applied further filtering criteria to remove potential genomic contaminations and rare PCR artifacts. An isoform was retained if the start and end sites fell within 50 bp of an annotated CAGE or poly-(A) peak from published CAGE peaks and polyA site data[83,84]. Additionally, isoforms that contained novel non-canonical or RT-template switching junctions will be excluded[44]. We further analyzed AS events using SUPPA2 (v 2.3)[85] with the following parameters "-f ioe -e SE SS MX RI FL" for each cell type.

**Generation of single-cell gene count matrices and cell clustering of the sequencing data from PacBio.** We generate a collated.CSV file using the Cupcake/singlecell scripts that linked each mapped FLNC read to its classified genes and transcripts based on the filtered GTF file from SQANTI3's output. We then used make_seurat_input.py to generate a Seurat-compatible input based on SQANTI3's output and the collated.CSV file using the Cupcake/singlecell scripts. This script makes a sparse matrix based on the isoform counts per cell. This output matrix was loaded into Seurat for the standard Seurat pipeline using the following parameters: "min.cells=3, min.features=50". Clustering was performed with a resolution of 0.1. Integration of the Illumina short-read matrix and PacBio long-read isoform matrix was carried out with the LIGER package (v 1.0.0) in R (https://github.com/JEFworks/liger). The correlation of the scRNA-seq and Iso-seq date was analyzed in R (https://www.r-bloggers.com/2021/01/correlation-analysis-in-r-part-2-performing-and-reporting-correlation-analysis/).

## Bulk RNA sequencing and analysis

Total RNA extracted from P26 – 30 mouse cochlea was used for library construction and deep sequencing using BGISEQ-500 (BGI-Shenzhen, China). Clean reads were then mapped to the mouse reference genome (mm10) using STAR (v2.7.9a) to detect junction sites[86], using the following options: '--outFilterMultimapNmax 1 --outFilterIntronMotifs RemoveNoncanonical --outFilterMismatchNmax 5 --alignSJDBoverhangMin 6 --alignSJoverhangMin 6 --outFilterType BySJout --alignIntronMin 25 --alignIntronMax 1000000 --outSAMstrandField intronMotif --outSAMunmapped Within --outStd SAM --alignMatesGapMax 1000000'. Junctions supported with more than 5 unique reads were retained and annotated as known or novel based on the Gencode database.

Previously published data from mouse left cerebral cortex tissue and the layer of hippocampus tissue (ENCODE: ENCSR248KDJ and ENCSR330DDD) were mapped to the mouse reference genome (mm10) using STAR. Bigwig tracks for visualization were produced using DeepTools. All genome browser screenshots and sashimi plots were produced using IGV (v 2.11.1)[87].

## Oxford Nanopore sequencing data analysis

Total RNA was extracted from the mouse organ of Corti (P26–30) using TRIzol reagent (TIANGEN BIOTECH, China). The integrity of the RNA was determined with an Agilent 2100 Bioanalyzer (Agilent Technologies, USA) and agarose gel electrophoresis. We used a Nanodrop (Thermo Fisher Scientific, USA) and Qubit (Thermo Fisher Scientific, USA) to determine the purity and concentration of RNA samples. Only high-quality RNA samples (OD260/280 = 1.8 - 2.2, OD260/230 ≥ 2.0, and a RIN ≥ 8) were used to construct sequencing libraries. Total RNA was reverse transcribed to cDNA using the SQK-PCS109 cDNA-PCR sequencing Kit (Oxford Nanopore Technologies) with Maxima H Minus Reverse Transcriptase (CAT#EP0751, Thermo Fisher Scientific, USA) and then amplified using suitable PCR cycles with using LongAmp Taq 2X Master Mix (CAT#M0287, NEB). Damage repair and end repair were carried out before cDNA was ligated to sequencing adapters using the SQK-LSK109 Ligation Kit (Oxford Nanopore Technologies) and purified using Agencourt XP beads (Beckman Coulter, USA). Finally, EXACT NUMBER libraries were constructed and sequenced on EXACT NUMBER different R9.4.1 FlowCells using a PromethION sequencer at Grandomics Biosciences (Beijing, China).

Full-length reads were identified and oriented using the pychopper tool (https://github.com/nanoporetech/pychopper) with default parameters and then were aligned to the mouse reference genome (mm10) using minimap2 (-ax splice -uf --junc-bed). To obtain transcript isoform sequences, clustering of full-length reads and redundancy removal were performed using StringTie2 software (v 2.1.4, https://ccb.jhu.edu/software/stringtie), and then the isoform sequences were corrected using the reference genome sequence. We utilized GffCompare (v 0.12.6) (https://github.com/gpertea/gffcompare) to examine the similarities and differences between sequence sample gene isoforms and reference gene annotations.

## RNA extraction, RT–PCR, and quantitative PCR (qPCR)

The cochlear basilar membrane from P26–30 mouse inner ears was carefully dissected in RNAlater (Thermo Fisher Scientific, USA) or phosphate-buffered saline (PBS), pH 7.2 to 7.4, and then stored in RNAlater. RNA was extracted using QIAshredder columns (Qiagen, Germany) and the RNeasy Mini Kit (Qiagen, Germany) following the manufacturer's instructions. The RNA concentration was measured using a NanoDrop spectrophotometer (Thermo Fisher Scientific, USA). cDNA was prepared using the PrimeScript™ RT reagent Kit with gDNA Eraser (Takara, China). The primer sequences used are designed by ourselves using SnapGene software (v 4.2.4) and synthesized by Sangon Biotech (Shanghai, China) and listed in Supplementary Data 5. Sanger sequencing was generated by ABI 3730 xl platform (Thermo Fisher Scientific, USA), and the data was analyzed using SnapGene software (v 4.2.4). Quantitative real-time PCR (qPCR) was performed using an ABI Real-Time PCR Detection System with SYBR Green qPCR Master Mix (Thermo Fisher Scientific, USA).

## Mass spectrometry-based proteomics and data analysis

For evaluation of the isoforms that could translate into proteins, the organ of Corti from P26–30 wild-type mice were treated with ice-cold RIPA lysis buffer plus a protease inhibitor cocktail (Thermo Fisher Scientific, USA) and phosphatase inhibitors (Thermo Fisher Scientific, USA). In total, 200 μg of protein was used for MS sample preparation. After reduction (with dithiothreitol, DTT) and alkylation (with iodoacetic acid, IAA), proteins were digested using trypsin (1:40 (w/w)) at 37 °C overnight. Peptides were then fractionated by basic reversed-phase liquid chromatography using an Ultimate 3000 system (Thermo Fisher Scientific, USA) connected to a reverse phase column (XBridge C18 column, 4.6 mm × 250 mm, 5 μm, (Waters Corporation, USA). Twenty-four fractions were collected, and each fraction was subsequently dried using a vacuum concentrator for the next step. All peptides were then analyzed by online nanoflow liquid chromatography-tandem mass spectrometry performed using an EASY-nanoLC 1200 system (Thermo Fisher Scientific, USA) connected to an Orbitrap Fusion™ Lumos™ Tribrid™ mass spectrometer (Thermo Fisher Scientific, USA). An Acclaim PepMap C18 (75 μm ×25 cm) column was equilibrated with solvent A (A: 0.1% formic acid in water) and solvent B (B: 0.1% formic acid in 80% ACN). The appropriate peptides were loaded and separated with a 60 min gradient at a flow rate of 400 nl/min. The mass spectrometer was run in the data-independent acquisition (DIA) mode and automatically switched between MS and MS/MS modes. Tandem mass spectra were processed using Spectronaut (v 14, Biognosys, Schlieren, Switzerland) with default settings. Spectronaut was set up to search our ScISOr-Seq predicted ORFs database and/or the UniProt database.

## Generation of knockout mice

*Otof-ΔC* mice: The genotypic animal was designed and developed by Shanghai Model Organisms Center, Inc (Shanghai, China). Briefly, Cas9 mRNA was in vitro transcribed with mMACHINE T7 Ultra Kit (Invitrogen, USA, no: AMB13455) according to the manufacturer's instructions. Two guide RNAs (gRNAs, synthesized by Sangon Biotech (Shanghai, China)) targeted to delete exon 2 were in vitro transcribed using the MEGAscript T7 Kit (Invitrogen, USA, no: AM1354). The gRNA sequences were 5′-CAGAACAGCATTTGACCCTTTGG and 5′-TGCTGCACA-CAAGCCACCCAGGG. These in vitro transcribed gRNAs were purified using the NucleoSpin miRNA Kit (Macherey-Nagel, Germany) and quantified using a NanoDrop spectrophotometer. A mixture containing each gRNA at 30 ng/μl and 110 ng/μl Cas9 mRNA (Trilink Biotechnologies, USA) was injected into zygotes and implanted into pseudopregnant females. Founders were screened for mutations in targeted exons as well as for large deletions in intervening sequences, backcrossed to C57BL/6 J mice, and propagated for the experiments described in this study. *Otof-ΔC* (*Otof* ^exon2-/-^) mice were confirmed by sequencing the mouse tail genomic DNA with the following primers: 5′-GCCTGGGCTTTCCTACATTTTA-3′ and 5′-AACCCCGCTGACCCCTTA-GAGA-3′ (synthesized by Sangon Biotech (Shanghai, China)). The PCR amplicon sizes were 577 bp and 344 bp for WT mice and *Otof- ΔC* mice, respectively. *Otof-ΔS* mice: The gRNAs designed based on the deletion of exon 6b (5′-CCTCACCTGCCTCTAGACGT; 5′-AACAGGTACATAGGT CCGGG). Mice were genotyped using the following primers: 5′-AACATGACAGGAGCAGTCCCTGT-3′ and 5′-CCCAAAGGACATAGGA-CACTTCTTC-3′. The PCR amplicon sizes were 1022 bp (WT) and 511 bp (*Otof- ΔS*). *Otof* full KO mice (denoted *Otof* ^-/-^): otoferlin full KO mice were obtained by deletion of exons 14 and 15 as previously reported[25]. *Otof* ^-/-^ mice were genotyped using the following primers: 5′-CACCGGGGTTAATGGAAGGT-3′ and 5′-TGTCTCCCTGTGTTCCTGG

AT-3′. The following PCR products were generated: a band of 1766 bp for WT (*Otof*$^{+/+}$) mice and a single 693 bp product in homozygous (*Otof*$^{-/-}$) mice.

### Hearing assessment and acoustic exposure

Mice were anesthetized with a combination intraperitoneal injection of ketamine (30 mg/kg; Sigma-Aldrich, USA) and pentobarbital (50 mg/kg; Sigma-Aldrich, USA). ABR recordings were made in a sound-attenuating chamber. A TDT RZ6 workstation (Tucker–Davis Technologies, USA) was used to deliver acoustic stimuli and record the response signal. Short tone burst stimuli (3 ms duration, 1 ms rise/fall times) were delivered free field through an MF-1 speaker, which was positioned 10 cm away from the vertex at a rate of 20 per second. Stimulus frequencies rove from 32 to 4 kHz were presented in half-octave steps. For each measured frequency, the sound level started from 90 dB SPL and descended in 5 dB steps until two levels below visible thresholds. Each waveform was averaged 400 times. The analysis of the latency (ms) and amplitudes (μV) of ABR wave I were carried out offline using BioSigRZ software (v 5.1, Tucker-Davis Technologies, USA).

Noise exposure was performed in a calibrated reverberating chamber where differences in the sound pressure level varied by ~1 dB in typical locations. Noise signals were generated using a TDT RZ6 system (Tucker-Davis Technologies, USA) and calibrated to the target sound pressure level immediately before each acoustic over-exposure by an acoustimeter (Hangzhou Aihua, China). A bandpass noise of 2–20 kHz at 103 dB SPL was delivered for 2 h using an amplifier and loudspeaker (Yamaha, Japan).

### Immunohistochemistry and confocal microscopy

The cochleae from P26-30 mice were perfused with 4% paraformaldehyde immediately after dissection and fixed for 30 min at room temperature. The tissue was decalcified in 10% EDTA (Sigma-Aldrich, Germany) for 2 h, and the organ of Corti was microdissected for whole-mount imaging. Thereafter, specimens were permeabilized and blocked for 60 min in 0.5% (v/v) Triton X-100 and 4% (w/v) BSA/PBS at room temperature before incubation with the following primary antibodies: mouse anti-CtBP2 IgG1 (BD Biosciences, USA, no: 612044, dilution:1:200), Mouse anti-GluR2 IgG2a (Merck-Millipore, Germany, no: MAB397, dilution:1:200), mouse anti-otoferlin (Abcam, USA, no: ab53233, dilution:1:200), rabbit anti-otoferlin (Synaptic Systems, Germany, no: 178003, dilution:1:200), mouse anti- Myosin-VIIa DSHB, USA, no: 138-1, dilution:1:200), and rabbit anti-Endophilin 1 (Synaptic Systems, Germany, no: 159002, dilution:1:200). The secondary antibodies used were as follows: Alexa Fluor 568-conjugated goat anti-mouse IgG1 (Invitrogen, USA, no: A-21124, dilution:1:500), Alexa Fluor 647-conjugated goat anti-mouse IgG2a (Invitrogen, USA, no: A-21241, dilution:1:500), and Alexa Fluor 488-conjugated goat anti-rabbit IgG (Invitrogen, USA, no: A-11008, dilution:1:500). Finally, DAPI (Invitrogen, USA, no: P36941, 20 μl) was pipetted onto each slide for 10 min at room temperature. Confocal images were acquired using Zeiss LSM 880 with Zen imaging software (v2.3) and processed using Imaris software (v 9.2.0).

### Patch-clamp recordings from IHCs

All IHC recordings were performed in the apical turn of the organs of Corti from P26–30 mice using an EPC10/2 amplifier (HEKA Electronics, Germany) driven by Patchmaster software (HEKA Electronics, Germany). Patch pipettes were pulled from borosilicate glass capillaries (World Precision Instruments, USA) using a vertical puller (PC-100, Narishige, Japan) to obtain a resistance range of 5-6 mΩ. The tissue was placed in a recording chamber perfused with the extracellular solution containing: 110 mM sodium chloride, 2.8 mM potassium chloride, 25 mM tetraethylammonium chloride, 5 mM calcium chloride, 1 mM magnesium chloride, 2 mM sodium pyruvate, 5.6 mM D-glucose and 10 mM 4-(2-hydroxyethyl)-1-piperazineethanesulfonic acid (300 mOsm, pH 7.40). Pipettes were coated with dental wax and filled with an intracellular solution containing the following: 120 mM cesium methanesulfonate, 10 mM cesium chloride, 10 mM 4-(2-hydroxyethyl)-1-piper-azineethanesulfonic acid, 10 mM tetraethylammonium chloride, 1 mM ethylene glycol-bis (β-aminoethyl ether)-N, N, N′, N′-tetraacetic acid, 3 mM adenosine triphosphate magnesium, and 0.5 mM guanosine 5′-triphosphate sodium salt hydrate (pH ~7.20, 290 ~mOsm). All chemicals were purchased from Sigma-Aldrich (USA). Recordings were discarded if the leak current exceeded −50 pA at a −90 mV holding potential. All patch-clamp experiments were performed at room temperature, and the liquid junction potential was corrected offline.

Whole-cell membrane capacitance ($C_m$) measurements in IHCs were performed with the lock-in feature and the "Sine+DC" method in Patchmaster software. Briefly, a 1 kHz sine wave and 70 mV peak-to-peak magnitude were superposed on the IHC holding potential of −90 mV. The increase in $C_m$ ($\Delta C_m$) after membrane depolarization was used to monitor exocytosis from IHCs, and the $Ca^{2+}$ charge ($Q_{Ca}$) was calculated by taking the integral of the leak-subtracted current during depolarization.

### Transmission electron microscopy

The organ of Corti from P26-30 mice were dissected and fixed with 2.5% glutaraldehyde in phosphate buffer (pH 7.2) overnight at 4 °C. The samples were then washed with 0.1 M sodium cacodylate buffer and placed in 1% osmium tetroxide (v/v in 0.1 M sodium cacodylate buffer) for 2 h at room temperature in the dark. Next, the samples were washed twice in 0.1 M sodium cacodylate buffer (10 min each, on ice), further washed in distilled water, and subsequently stained en bloc with 1% uranyl acetate (v/v in distilled water) for 1 h on ice. Uranyl acetate-treated samples were briefly washed three times in distilled water, dehydrated through a series of increasing ethanol concentrations, and finally embedded in epoxy resin (Agar 100 kit, Plano, Germany) for polymerization for 48 h at 60 °C.

An Ultracut E microtome (Leica Microsystems, Germany) equipped with a 35° diamond knife (Diatome, Switzerland) was used to obtain ultrathin sections (60–80 nm) of the specimen. The sections were transferred to 1% formvar-coated (w/v in water-free chloroform) copper slot grids (ATHENE copper slot grids, 3.05 mm Ø, 1 mm × 2 mm; Plano, Germany) and subsequently stained with uranyl acetate replacement solution (UAR-EMS) (Science Services, Germany) and Reynold's lead citrate. The specimens were investigated at 80 kV with a JEM1011 transmission electron microscope (JEOL, Germany), and micrographs were acquired at ×10,000 magnification using a Gatan Orius 1200 A camera (Gatan, Germany), using the Digital Micrograph software package (v 01.12). Ribbon-associated SVs (RA-SVs) were located in the first row around the synaptic ribbon. Cytosolic SVs were vesicles found in the 1 μm radius region around the ribbon, excluding RA-SVs. Coated vesicles were defined as clathrin-coated structures with a maximum diameter less than 70 nm. Coated vacuoles were clathrin-coated structures with a maximum diameter larger than 70 nm. ELVs were also defined as structures with a maximum diameter larger than 100 nm.

### Protein binding assays

**Plasmids construction.** All cDNA templates were synthesized to pUC57 or pClone007 by Sangon Biotech (Shanghai, China). cDNAs were then cloned into plasmids by inserted into pcDNA3.1-3xFlag between BamHI and XhoI sites, into pcDNA3.0-UTR-HA between EcoRI and XhoI sites, into pET28a between BamHI and XhoI sites, into pGEX6p-1 between EcoRI and SalI sites with homologous recombination enzyme (Vazyme, China). In this work, the template of canonical mouse Otof (1-1885 AA) was produced, while the short mouse Otof used in this article was produced based on that canonical one without the first 192 AA. All other templates used canonical isoform. All

constructs were verified by DNA sequencing. All plasmids used in this study were: pcDNA3.0-UTR-*Otof-C-HA*, pcDNA3.0-UTR-*Otof-S-HA*, pcDNA3.0-*Otof-C-Egfp*, pcDNA3.0-*Otof-S-Egfp*, pET28a-*Otof-C₂ABC* (coding: 1-616 AA), pET28a-*Otof -C₂A* (coding: 1-192 AA), pET28a-*Otof -C₂BC* (coding: 193-616 AA), pGEX6P-1-*Otof -C₂ABC*, pGEX6P-1-*Otof -C₂A*, pGEX6P-1-*Otof -C₂BC*, pGEX6P-1-*Sh3gl2*, pGEX6P-1-*Clta*.

**Cell culture and plasmids transfection.** HEK-293T (ATCC) was cultured in DMEM (Corning, USA) supplemented with 10% fetal bovine serum (Lonsera, China) and 100 units/ml penicillin and 100 ug/ml streptomycin (Beyotime, China). Cells were incubated at 37 °C with 5% $CO_2$. All transfection experiments were performed using Lipofectamine 2000 (Invitrogen, USA) or polyethyleneimine (Sigma, USA), following the manufacturer's instructions.

**Co-immunoprecipitation and Immunoblotting assay.** For co-immunoprecipitation, cells expressing the proteins of interest were lysed in co-IP buffer (50 Mm Tris-Cl PH 7.6, 150 mM NaCl, 5.0 mM EDTA, 1.0% NP-40) supplemented with protease inhibitor cocktail (Roche, Switzerland) and sonicated using the Vibra-Cell processors. After centrifuged at 22,500 g at 4°C for 15 min to remove the cell debris, the cleared supernatant lysates were then incubated with Anti-HA beads (GNI, Japan, no: GNI4510-HA, 15 μl) overnight at 4 °C. The beads were pelleted and washed five times with co-IP buffer before the Immunoblotting assay. For immunoblotting, samples were denatured at 100 °C for 15 min in SDS-PAGE loading buffer and then subjected to SDS-PAGE before blotting to PVDF or NC membranes (Bio-Rad, USA). Membranes were then blocked with 10% non-fat milk, incubated with specific primary antibodies and secondary antibodies, finally visualized with ECL Western Blotting Reagent (Tanon, China). The primary antibodies include mouse anti-GFP tag (Proteintech, USA, no: 66002-1-Ig, dilution:1:2000), rabbit anti-HA-Tag (Cell Signaling Technology, USA, no: 3724 T, dilution: 1:1000), mouse anti- 6xHis-tag (Sigma, USA, no: H1029, dilution: 1:3000), and mouse anti-GST (Santa Cruz Biotechnology, USA, no: sc-138, dilution: 1:500). The secondary antibodies include Peroxidase AffiniPure Goat Anti-Mouse IgG (H+L) (Jackson ImmunoResearch, USA, no: 115-035-003, dilution: 1:10000) and Peroxidase AffiniPure Goat Anti-Rabbit IgG (H+L) (Jackson ImmunoResearch, USA, no: 115-035-003, dilution: 1:10000).

**Expression and purification of recombinant proteins.** GST or His6-tag protein plasmids were expressed in the BL21(DE3) or ROSET-TA(DE3) strains and grown firstly at 37 ˚C in LB medium supplemented with appropriate antibiotics. When OD600 reached 0.6–0.8, protein expression was then induced at 16 ˚C with 0.5 mM IPTG for more than 16 h. To purify His6-tagged proteins, bacteria were harvested and lysed in Ni-NTA Binding buffer (50 mM Na2HPO4, 500 mM NaCl, 20 mM imidazole, 0.1% TritonX-100, 1 mM DTT). Proteins were purified with affinity chromatography using Ni-NTA beads (GE) according to the manufacturer's instruction with wash buffer (50 mM Na2HPO4, 500 mM NaCl, 40 mM imidazole, 0.1% TritonX-100, 1 mM DTT) and elution buffer (50 mM Na2HPO4, 300 mM NaCl, 250 mM imidazole). For GST-fusion proteins, bacteria were lysed in NETN buffer (50 mM Tris-Cl pH 7.5, 150 mM NaCl, 0.5% NP-40, 1 mM EDTA, 1 mM DTT). Purifications were performed by affinity chromatography using Glutathione Sepharose Fast Flow beads (Sangon Biotech, China). Beads were washed with NETN buffer and then eluted by elution buffer (200 mM Tris-Cl pH 8.0, 20 mM Glutathione). All eluted proteins were further dialyzed in PBS overnight at 4 ˚C. Recombinant proteins were concentrated and then stored at −80 ˚C. Protein concentrations were determined using Bradford colorimetric assays, with their protein purities examined with SDS-PAGE followed by Coomassie Blue staining.

**GST pull-down assay.** Purified GST or His6-tag proteins for GST pull-down assay were incubated in GST pull-down buffer (20 mM Tris-Cl, 100 mM NaCl, 5 mM MgCl2, 1 mM EDTA, 1 mM DTT, 0.5% (v/v) NP-40, PH 7.6) with Glutathione Sepharose 4B (GE) for 4 h at 4 °C. The beads were washed 5 times with GST pull-down buffer and analyzed by immunoblotting assay. Buffers were adjusted when pull-down assay including calcium ion (20 mM Tris-Cl, 100 mM NaCl, PH 7.6 in different calcium concentrations).

**Statistical analyses.** Igor Pro (WaveMetrics Inc., USA) and GraphPad Prism (GraphPad Software Inc., USA) were used for statistical analysis. Depending on the nature of the dataset, statistical significance was assessed using a two-tailed unpaired Student's *t* test, Mann–Whitney test, or one- or two-way ANOVA, followed by a Bonferroni *post hoc* test. Results are shown as mean ± SD, and the level of significance was set to $p < 0.05$. Significant differences were reported as $^{*}p < 0.05$, $^{**}p < 0.01$, and $^{***}p < 0.001$.

### Reporting summary
Further information on research design is available in the Nature Portfolio Reporting Summary linked to this article.

## Data availability
The raw data of scRNA-seq, Iso-seq, and bulk RNA-seq generated in this study have been deposited in the NCBI Sequence Read Archive (SRA) under accession BioProject codes: PRJNA759047 (BioSample: SAMN21155185). Mass spectrometry-based proteomics data have been deposited at the ProteomeXchange Consortium via the PRIDE partner repository with the accession number PXD031062. RNA-seq data of mouse cerebral cortex tissue and hippocampus tissue are available in a public repository from the https://www.encodeproject.org/ website (ENCODE: ENCSR248KDJ and ENCSR330DDD). The reference mouse genome (GRCm38/mm10) was downloaded from the UCSC genome Browser (http://genome-asia.ucsc.edu/cgi-bin/hgGateway?redirect=manual&source=genome.ucsc.edu). scRNA-seq data can be accessed at https://umgear.org/index.html?multigene_plots=0&layout_id=0d6584b2&gene_symbol_exact_match=1&gene_symbol=ocm. The novel transcripts in different cell types can be accessed using the "UCSC Genes track" file at https://genome-asia.ucsc.edu/. The remaining data are provided with this paper within the Article, Supplementary Information, or Source Data file. Source data and the UCSC Genes track file are provided in this paper. Source Data are provided with this paper.

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

## Acknowledgements

We thank Dr. Wendol Williams for critical reading and editing the manuscript, Jingjing Li (Grandomics Biosciences Co., Ltd.) for the help of bioinformatics analysis, and Bioimaging Facility of the Shanghai Institute of Precision Medicine. This work was supported by research grants from the National Natural Science Foundation of China (82101211, 81970872) and an institutional research grant from Shanghai Science and Technology Commission to Shanghai Key Laboratory of Translational Medicine on Ear and Nose Diseases (14DZ2260300).

## Author contributions

H.W. conceived and designed the research. Hh.L. and Lh.W. performed omics' analysis and TEM experiments. Hh.L. and Hc.L. performed electrophysiological and morphological experiments. Hh.L., Q.L., T.Z and G.J. designed and performed protein binding experiments. L.S. and T.Y. provided valuable comments. Hh.L. and H.W analysis the data and wrote the manuscript. L.S., H.P., RJ.C. and Xl.Z. reviewed and edited the manuscript.

## Competing interests

The authors declare no competing interests.
