## [Peer Review File · Nature Communications]

Cochlear transcript diversity and its role in auditory functions implied by an otoferlin short isoformREVIEWER COMMENTS

Reviewer #1 (Remarks to the Author):

This study by Liu et al. addresses a significant gap in inner ear and hair cell research. To date, only a handful of studies have highlighted the functional significance of protein isoform in the auditory system, usually focusing on a single gene (Myo7a, Myo15, Whirlin, etc.). This group applied a powerful combination of transcriptomics (10x Illumina and PacBio long reads) and proteomics approaches to investigate the landscape of gene isoforms in the cochlea in a systematic manner. This study, if the data is made available in a meaningful manner, will be impactful in guiding future studies on protein isoforms and their relevance for inner ear related diseases. Overall, this study suggests that most genes are expressed in multiple isoforms, opening up a vast area of new research directions to study their functional relevance.

In the latter part of the paper, the authors focus on the protein otoferlin, to demonstrate the functional significance of a novel protein isoform identified in their initial big data approach. The authors claim that their study of otoferlin isoforms, and isoform-specific KO mice, provides a new understanding of OTOF's biology in presynaptic vesicle recycling, potentially establishing a new model for hidden hearing loss found in their isoform-specific KO mice.

While the first part of the paper is well executed and yields data that is undoubtedly useful for the field, the latter part, focusing on OTOF, contains overinterpretations and shortcomings that requires major re-writes and some additional experiments to justify publication at this level. The main concern is that this reviewer is not convinced that the two isoforms (OTOF-C and -S) indeed fulfill distinct function as claimed by the authors, and that their findings are somewhat inconsistent with previous work on OTOF knockdown and knockout mice.

Major comments:

1. The authors appear to claim that the two isoforms fulfill two distinct functions. Yes, due to the lack of one domain, the newly identified, shorter isoform interacts to a lesser degree with pre-synaptic proteins (as shown by Co-IP), but a definitive connection to the phenotype observed in the isoform specific KO mice is missing. Most aspects of the phenotype can be explained by a gene dosage effect. The authors claim in Line 375 that gene dose effects can be excluded "because of the two heterozygous mutant mice lines showed normal auditory brainstem responses". It should be noted however, that heterozygote mice can have protein levels similar to WT levels, and a previous publication appears to indicate that in the case of OTOF (see Fig. 6 in Pangrsic et al 2010).
2. There is a significant discrepancy between the conclusions of this paper with previous ones. For example, the Pga mouse from Pangrsic et al (2010) exhibits no ABR at all, yet, they did not detect any difference in vesicle endocytosis, while the Otof-DeltaC mice of this study has near perfect ABRs, but has a defect in endocytosis. The authors make some effort in explaining this, but it was not entirely convincing.
3. According to supplementary figure 4C, the expression level of OTOF-C and OTOF-S are similar. However, in figure 4A-C, at protein level. the deletion of the OTOF-C does not cause an obvious reduction of the OTO-C signal. This needs to be quantified, best normalized to some hair cell specific

gene like Myo7a or Myo6.

4. All claims regarding intensity differences made in Figure 7 are shaky. The n for statistical analysis in all cases is “spots”, but should at least be “cell”, if not “organ” or “animal”. The intensity quantification data in Fig.7C (tonotopic differences) and especially Fig.7G (and J) needs to be reanalyzed using cell, organ or animal as n. It is quite likely that the claimed reduction of OTOF levels shown in Fig 7G and J will not hold up (no significant difference) when proper stats are used. In that case, this piece of the manuscript could be omitted without losing too much impact for the paper as a whole.

5. This reviewer represents an audience who is less familiar with mining large datasets. If it is the intention of the authors to make this data readily available, it would be helpful to have a tutorial on how the data can be accessed. For example, if one is interested in isoforms of a specific gene in a specific cell type, how that analysis could be achieved.

Other comments:

1. Figure 2:

a) G: Although the same color code is used in figure 1, it would be better to have color legends on the side of each figure.

2. Figure 3:

a) C: The y-axis does not range from 0~100%, and 0 is not aligned with the bar plot.

b) D: Upper part of ‘E1b’ cut off

c) F, G: Add a legend for the color code.

3. Figure 4:

a) A, B, and C:

i. The image does not represent the entire length of the cochlea. Why is the basal part missing?

ii. Please remove the yellow “Merge” annotation; it is not a color used in either of those channels. And please indicate which is the staining of the blue channel.

iii. Please add error bars and specify the age of the mouse in the figure legend.

4. Line 349~350 “Our histological analysis revealed that these antibodies did not distinguish isoforms in wild-type (WT) IHCs.” This wording is confusing. Do the authors mean that the antibody can distinguish these two isoforms, but expression patterns of these two isoforms did not show a substantial difference?

5. Line 365-367, “The ABR threshold reports the function of OHCs that support the sensitivity of auditory sensing. Thus, the OTOF-ΔC mice had normal OHC function”:

To evaluate the function of OHCs, DPOAE should be tested on these mice.

6. Figure 6:

a) B, E: Please indicate the percentage of the input used in the input gel.

b) C: The author did not specify which value the fluorescent signal is normalized to.

7. Line 447, “except for the N terminal C2A region, the structure of the shorter alternatively spliced isoform superimposes well on the structure of the conventional one...”:

The two isoforms are identical with the exception of the N-terminal addition. The AlphaFold2 analysis in this case thus is trivial and not very informative and could be removed.

8. Line 463~467, “C2B-F has the ability to bind to under different Ca²⁺ concentrations”:

This sentence is confusing. Please consider rephrasing it.

9. Line 471~473, "Collectively, these results suggested ... biological functional difference."

There is no significant reduction of endophilin A1 colocalization in OTOF-deltaC mice in figure 6C, which weakens the conclusion that there indeed is a biological functional difference.

10. Figure 7:

a) A: Scale bar. Legend for the blue channel. Remove the yellow "Merge" legend.

b) C, G, J: Statistical analysis should be conducted on the number of organs or the animals (especially for panel G).

6. In general, for the different gene transcripts, what is the average abundance of novel transcript of each gene? This information might be challenging to obtain, but if there is a way, it would be very helpful to identify the (presumably more abundant) isoforms with functional importance.

7. Is it possible, using the SQANTI classification, to distinguish between transcripts that are generated by alternative start sites vs alternative splicing events? This will help to answer if these transcripts are regulated at the gene expression level or splicing level.

8. The paragraph about the Actb isoforms (lines 218-227) is difficult to understand. Please rephrase.

Reviewer #2 (Remarks to the Author):

Transcriptomic analyses have been extensively employed to investigate the transcription diversity in the cochlea. However, much less is known about the alternative splicing events in the cochlea at the transcriptomic level. In the present work, the authors combined single-cell short-read and long-read RNA sequencing techniques to identify the alternative splicing events in the cochlea. They then focused on Otoferlin and showed that different Otoferlin isoforms are associated with distinct IHC functions using a knockout mouse model. This study has great significance and broad interest in the related field. Some concerns and suggestions are listed below.

Major points:

1. It's important to confirm the two Otoferlin isoforms at the protein level. Can the authors detect the two isoforms in the cochlea by performing western blot?

2. The authors deleted the exon 2 of Otoferlin gene and obtained knockout mice that only express the novel isoform but lack the canonical isoform. The knockout mice have normal ABR threshold but reduced wave I amplitude. Further investigation revealed that the knockout mice show reduced capability of sustained exocytosis. These results are very interesting and helpful for understanding the role of Otoferlin in the ribbon synapse. However, to investigate the role of the novel Otoferlin isoform (OTOF-S), a more straightforward strategy would be deleting exon 6b and establishing knockout mice without functional OTOF-S.

Minor points:

1. Line 47: "... without no noticeable..." should be "... and no noticeable...".

2. Line 321: The authors declared that the novel Otoferlin transcript is "a novel inner ear-specific transcript". It will be nice if they can validate this tissue-specific pattern by performing RT-PCR in

different mouse tissues.

3. Line 635: The sentence “The coexisting isoforms cannot...” is confusing to the reviewer. I suggest using “the novel isoform” instead of “the coexisting isoforms”.

4. The language needs to be improved.

Reviewer #3 (Remarks to the Author):

In this manuscript, Liu et al. used a combination of single cell- short- and long-read sequencing to explore the transcriptomic landscape of the cochlea. They complemented their study with a mass spectrometry-based proteomics to confirm their findings. They also focused on one of the novel transcripts they identified for OTOF gene; a short isoform lacking the C2A domain. Their KO mouse models showed distinct functions and expression patterns tonotopically for each isoform and suggests that the canonical isoform might be involved in hidden hearing loss.

This work provides a wealth of data and valuable insights into the transcriptomic landscape of different cell types in the cochlea. This is a much-needed knowledge that will enable research in all fields of auditory sciences. The characterization of a novel short isoform for OTOF that has a distinct function is extraordinary and has direct translational consequences for the ongoing clinical trial as well as for clinical genetic testing, natural history studies and understanding of the mechanisms involved in hidden hearing loss. Nevertheless, several issues need further clarification and correction.

Major comments:

1-To take full advantage of the data in this work, IHC and OHC should have been placed in two distinct clusters. We do know 1-their morphology and function are different, and 2- most deafness-associated genes are expressed in these cells, so it is surprising their data were not analyzed separately to gain valuable insights into their physiology.

2-To investigate the role of the novel transcript Otof-S, an indirect method was used by creating a KO mouse model for the canonical isoform Otof-C. Is there a reason why a KO of otof-S was not made instead or in addition? I think creating a KO for the otof-S is more valuable and will confirm the role of both the canonical and short isoforms.

3-A previous work by Ranum et al., (Ranum PT, Goodwin AT, Yoshimura H, Kolbe DL, Walls WD, Koh JY, He DZZ, Smith RJH. Insights into the Biology of Hearing and Deafness Revealed by Single-Cell RNA Sequencing. Cell Rep. 2019 Mar 12;26(11):3160-3171.e3. doi: 10.1016/j.celrep.2019.02.053. PMID: 30865901; PMCID: PMC6424336.) used a very similar approach combining short- and long-read sequencing to draw a picture of the inner ear transcriptome. However, there was no mention or reference to that work in this manuscript. Please reference the Ranum et al., paper and discuss your findings as compared to theirs.

4-The wealth of data provided by this work is hard to assess using the files included in the manuscript as the comprehensive aspect is lost. I strongly believe, this data need to be made available on a website similar to what was done by Ranum et al., (<https://morlscrnaseq.org/> and https://morlscrnaseq.org/JBrowse-1.12.3/?loc=5%3A142902388..142907483&tracks=mm10_gff%2CIHC_bigwig%2CIHC_SashimiPlot&highlight=) where users could easily access and visualize the different transcripts and data supporting them. Another way to make this data accessible to the greater community is to share it on the gEAR portal (<https://umgear.org/>) or to add it as a track on the genome browser (<https://genome.ucsc.edu/>).

5-Several gene therapies have been developed for OTOF-related hearing loss and a clinical trial is ongoing. The relevance of this work and its translational aspect need to be discussed. would the short isoform alone rescue hearing? Is there a need for dual vectors use for gene therapy for OTOF? Etc...

6-The number of cells isolated for each cell type as well as the methodology used to assign them to a specific cluster are not reported. This is very relevant to the interpretation of the data and needs to be added.

Minor Comments:

1-The manuscript needs to be reviewed for conciseness (Some sentences are 5-6 lines long). It will also benefit from a revision of syntax and grammar in some sections.

2-It would be great to add a summary figure to show the proposed tonotopic expression and distinct roles of Otof-C and Otof-S in IHC.

3-Figures: Please change figures' color palette to make them color-blind friendly

4-Fig2B and 3F: please add the abbreviations for isoform type to the pie chart itself. It makes it easier to understand.

5-Fig2F and 3G: please plot cell types on the pie chart. Avoid putting numbers inside the pie charts.

6-Line 365: "The ABR threshold reports the function of OHCs..." This is not accurate. DPOAE report the function of OHC. ABR threshold reports both IHC and OHC.

7-Please make sure to provide more specifics and explain some of your statements when needed. For example: Line363 " This was distinct from previous studies in other otoferlin mutation models"

8-Lines 295-297: is there a table associated with this data?

9-Lines 286-292: Where there peptides identified in your MS-based proteomics that were absent in your ScISOSeq isoforms? Please indicate and discuss.

10-Line 231: change 5'/3'-donor to 5'/3'-donor or acceptor.

11-Line 299: "In addition, 81 ScISOSeq specific novel peptides were discovered that were absent from the UniProt database." It will be great to add a table listing these novel peptides and corresponding proteins/genes and what makes them novel.

Point-by-Point Response to NCOMMS-22-36208

Title: The landscape of transcript diversity in the cochlea and its role in auditory functions: implications of a novel otoferlin isoform

Reviewer #1 (Remarks to the Author):

This study by Liu et al. addresses a significant gap in inner ear and hair cell research. To date, only a handful of studies have highlighted the functional significance of protein isoform in the auditory system, usually focusing on a single gene (Myo7a, Myo15, Whirlin, etc.). This group applied a powerful combination of transcriptomics (10x Illumina and PacBio long reads) and proteomics approaches to investigate the landscape of gene isoforms in the cochlea in a systematic manner. This study, if the data is made available in a meaningful manner, will be impactful in guiding future studies on protein isoforms and their relevance for inner ear related diseases. Overall, this study suggests that most genes are expressed in multiple isoforms, opening up a vast area of new research directions to study their functional relevance.

In the latter part of the paper, the authors focus on the protein otoferlin, to demonstrate the functional significance of a novel protein isoform identified in their initial big data approach. The authors claim that their study of otoferlin isoforms, and isoform-specific KO mice, provides a new understanding of OTOF's biology in presynaptic vesicle recycling, potentially establishing a new model for hidden hearing loss found in their isoform-specific KO mice.

While the first part of the paper is well executed and yields data that is undoubtedly useful for the field, the latter part, focusing on OTOF, contains overinterpretations and shortcomings that requires major re-writes and some additional experiments to justify publication at this level. The main concern is that this reviewer is not convinced that the two isoforms (OTOF-C and -S) indeed fulfill distinct function as claimed by the authors, and that their findings are somewhat inconsistent with previous work on OTOF knockdown and knockout mice.

Response:

We thank the referee for the positive comments towards our work, and we are particularly grateful for the criticisms and suggestions that are critically helpful for improving the manuscript. We have modified the manuscript as suggested. Below, please find our response to each specific question.

Major comments:

1. The authors appear to claim that the two isoforms fulfill two distinct functions. Yes, due to the lack of one domain, the newly identified, shorter isoform interacts to a lesser degree with presynaptic proteins (as shown by Co-IP), but a definitive connection to the phenotype observed in the isoform specific KO mice is missing.

Most aspects of the phenotype can be explained by a gene dosage effect. The authors claim in Line 375 that gene dose effects can be excluded "because of the two heterozygous mutant mice lines showed normal auditory brainstem responses". It should be noted however, that heterozygote mice can have protein levels similar to WT levels, and a previous publication appears to indicate that in the case of OTOF (see Fig. 6 in Pangrsic et al 2010).

Response:

We thank the referee for raising this important point. First, **we apologize for confusing the reviewer in the original version that the two isoforms were functionally overlapped but did not possess distinct roles.**

Deleting exon 2 (*Otof*- ΔC mice) of *Otof* could produced an N-terminal truncate protein compared to the long isoform because the short (novel) isoform, of which the translation initiation region that starts at exon 6B, was not affected. Thus, only the short isoform was expressed in *Otof*- ΔC mice. Our results showed that the remaining short otoferlin isoform was still sufficient to maintain the normal ABR threshold but impaired vesicle recycling. However, in our additional experiments, the ABR and vesicle replenishment in the short isoform specific KO mice (*Otof* exon 6B deletion, ***Otof*- ΔS**) mouse were normal compared with WT mice. As a result, from the Co-IP experiment, we inferred that the electrophysiology phenotype of inner hair cells (IHCs) in *Otof*- ΔC

mice might be due to the shorter isoform interactions to a lesser degree with presynaptic proteins.

Therefore, we suggested that the **two otoferlin isoforms possessed approximate functions (functional overlap) not two distinct functions**, and the short isoform may act as a subsidiary role in maintaining the hearing function.

We thank the referee for pointing out this error. **We apologize for the wrong interpretations in our original manuscript. The phenotypes of *Otof*- ΔC mice were caused by deletion of the long isoform, and this can be explained by the gene dosage effect.** We intend to say that *Otof* is **not a haploinsufficient gene** and to interpret the *Otof*- ΔC mouse phenotype needs to consider the fact that this mouse line **is not a mere *Otof* hypomorph but null for a specific isoform (canonical isoform)**. We have corrected the wrong interpretations in our manuscript (**Line 386-388**).

To date, several different animal models have been used, such as *Otof*^{-/-}, *Otof*^{Pga/Pga}, *Otof*^{Ala515, Ala517/Ala515, Ala517} and *Otof*^{I515T/I515T} mice, and all these heterozygote mice showed normal hearing phenotypes¹⁻⁴. In addition, although the otoferlin expression level was decreased in heterozygote mice (*Otof*^{+/-}) (Fig. 6 in Pangrsic et al 2010, Supplementary Fig. 4c, d), no obvious hearing impairment was found. Moreover, humans with heterozygous mutation in the OTOF appeared to have a normal hearing function, as in the previous report⁵. Thus, the otoferlin gene was not a haploinsufficient gene.

The interpretation of the *Otof*- ΔC mouse phenotype needs to consider the fact that this mouse line is not a mere *Otof* hypomorph but null for a specific isoform. It is, therefore, possible that the loss of isoform-specific functions mainly causes some aspects of the described phenotype because the auditory function was normal in *Otof*- ΔS and *Otof*^{+/-} mice. In summary, the isoform-specific KO mouse models are complementary tools in studying otoferlin's role in hair cell function.

2. There is a significant discrepancy between the conclusions of this paper with previous ones. For example, the Pga mouse from Pangrsic et al (2010) exhibits no ABR

at all, yet, they did not detect any difference in vesicle endocytosis, while the *Otof-DeltaC* mice of this study has near perfect ABRs, but has a defect in endocytosis. The authors make some effort in explaining this, but it was not entirely convincing.

Response:

Thank you for raising this important question for us. **We apologize for not elaborating on the differences between *Otof-ΔC* mice and other models, and feel sorry for the confusion to this reviewer.** Otoferlin appears to operate exclusively in sensory hair cells and plays an essential but poorly understood role in controlling neurotransmitter release from the hair cell.

First, the generation methods of *Otof-ΔC* mice and *Otof^{Pga/Pga}* mice was different (**Response Fig.1**). The *Otof^{Pga/Pga}* mice carry a N-ethyl-N-nitrosourea-mediated *Otof* missense mutation leading to an amino acid substitution (Asp to Gly) close to the **C terminus** of the protein (see Fig. 3 F, Schwander, M. et al 2007)⁶. Therefore, this mutation could affect both **the canonical and the short (novel) isoform**, leading to lower protein levels of otoferlin expressed in IHCs (see Fig. 6 a-g, Pangrsic et al 2010), especially at the plasma membrane (see Supplementary Fig. 4, Pangrsic et al 2010), potentially together with minor structural changes and impairment of protein-protein interactions, leading to the auditory phenotype of *Otof^{Pga/Pga}*. The *Otof-ΔC* mice, on the other hand, express the novel isoform, and as demonstrated in our study, supports near normal auditory function.

Auditory brainstem response (ABR) is a non-invasive way to assess hearing performance. Wave I of ABR represents sound-evoked auditory nerve responses that correlate with the function of inner hair cell ribbon synapses. The amplitude of wave I of the ABR **represented the synchronous firing** of all the auditory nerve fibers and is highly correlated with inner hair cell synaptic functions.

Although *Otof^{Pga/Pga}* IHC could release synaptic vesicles and be capable of encoding sound into spiking activity from recoding a single auditory nerve fiber, the spikes of each auditory fiber may be desynchronization. In a typical ABR recording, ABR responses were averaged 400 times to achieve a good signal-to-noise ratio and **the desynchronized firing of the auditory nerve cannot summate to a**

distinguishable waveform, thereby explaining the absence of auditory evoked potentials of *Otof^{Pga/Pga}* mice. Our result of *Otof-ΔC* ABR demonstrated that upon exhaustive averaging, the amplitude of ABR wave were indeed decreased (Fig. 5d). In the *Otof-ΔC* mice model, we did find that some of the electrophysiological properties were similar to *Otof^{Pga/Pga}* mice, such as the decreased exocytosis and vesicle replenishment rate and declined ABR wave I with the increment of ABR averages (Fig. 5d) but overall the phenotypes were much milder.

For the endocytosis, the *Otof-ΔC* mice that expressed only the short isoform exhibited the impairment of endocytosis. Because the novel isoform lacks the C₂A domain, which could interact with synaptic proteins, such as endophilin⁷ and clathrin⁸, thus, we inferred that the canonical isoform could support the normal endocytosis while the short isoform was not. According to our findings, we inferred that the existence of the canonical isoform in *Otof^{Pga/Pga}* IHC could support the 'normal' vesicle retrieval, **although the vesicle release declined.**

In response to the reviewer's comment, we have expanded our interpretation of the *Otof-ΔC* mice phenotype in the discussion (Line 615-622).

Response Fig.1

Response Fig.1. The expression of otoferlin isoforms in different animal models. a The two category isoforms were expressed in the IHCs of WT mice. **b, c** *Otof-ΔC* mice and *Otof-ΔS* mice only express one isoform, the novel and the canonical

isoform, respectively. **d** The *Otof*^{Pga/Pga} mice could express novel and canonical isoforms, but the mutation affects both. **e** No otoferlin isoform was expressed in IHCs of otoferlin-ko mice.

3. According to supplementary figure 4C, the expression level of OTOF-C and OTOF-S are similar. However, in figure 4A-C, at protein level, the deletion of the OTOF-C does not cause an obvious reduction of the OTO-C signal. This needs to be quantified, best normalized to some hair cell specific gene like Myo7a or Myo6.

Response:

We thank the referee for the suggestion. In the revised manuscript, we included quantification in the relevant figures (Fig. 4a-d) as suggested.

Moreover, we also quantified the otoferlin expression level normalized to Myo7a (Supplementary Fig. 4 c, d). These results have been updated in our revised manuscript.

4. All claims regarding intensity differences made in Figure 7 are shaky. The n for statistical analysis in all cases is "spots", but should at least be "cell", if not "organ" or "animal". The intensity quantification data in Fig.7C (tonotopic differences) and especially Fig.7G (and J) needs to be reanalyzed using cell, organ or animal as n. It is quite likely that the claimed reduction of OTOF levels shown in Fig 7G and J will not hold up (no significant difference) when proper stats are used. In that case, this piece of the manuscript could be omitted without losing too much impact for the paper as a whole.

Response:

Thanks for the suggestion. We have now reanalyzed data based on the number of cells and annotated number of animals in the figure legend. To measure the OTOF-C immunofluorescence density ratio (not quantify the expression level), we normalized the fluorescence intensity of otoferlin puncta against a dark background⁹ surrounding the cell body and averaged per cell level changes (noted in the figure legend). Our results showed that the significant difference in the otoferlin isoform ratio (OTOF-C/(OTOF-C+OTOF-S)) remained along the tonotopic gradient after noise exposure and

aging.

We retained this part of the data to interpret whether, and to what extent, these isoforms participate in biological and pathological functions. We will be happy to further edit the text base on your comments.

5. This reviewer represents an audience who is less familiar with mining large datasets. If it is the intention of the authors to make this data readily available, it would be helpful to have a tutorial on how the data can be accessed. For example, if one is interested in isoforms of a specific gene in a specific cell type, how that analysis could be achieved.

Response:

Thank you for reminding us. We shared our scRNA-seq data on the gEAR portal (https://umgear.org/index.html?multigene_plots=0&layout_id=0d6584b2&gene_symbol_of_exact_match=1&gene_symbol=Ocm) and added a "UCSC Genes track" file in our revised manuscript to display how to view **novel isoforms** in a specific cell type (Response Fig.2).

Response Fig.2

Response Fig.2 Method for viewing the novel isoform. Users can upload the UCSC Genes track file (Supplementary Data 7) in the UCSC Genome Browser (<https://genome-asia.ucsc.edu/>). Once uploaded, custom track data can be manipulated

to view the novel isoforms in different cell types.

Other comments:

1. Figure 2:

a) G: Although the same color code is used in figure 1, it would be better to have color legends on the side of each figure.

Response:

Thanks for the suggestion. The color legend is now added in figure 2.

2. Figure 3:

a) C: The y-axis does not range from 0~100%, and 0 is not aligned with the bar plot.

b) D: Upper part of 'E1b' cut off

c) F, G: Add a legend for the color code.

Response:

Thanks for catching that! We have corrected these mistakes based on your suggestions.

3. Figure 4:

a) A, B, and C:

i. The image does not represent the entire length of the cochlea. Why is the basal part missing?

ii. Please remove the yellow "Merge" annotation; it is not a color used in either of those channels. And please indicate which is the staining of the blue channel.

iii. Please add error bars and specify the age of the mouse in the figure legend.

Response:

Thank you for providing the suggestions on figure organization. We have made the revisions according to your suggestion.

4. Line 349~350 "Our histological analysis revealed that these antibodies did not distinguish isoforms in wild-type (WT) IHCs." This wording is confusing. Do the

authors mean that the antibody can distinguish these two isoforms, but expression patterns of these two isoforms did not show a substantial difference?

Response:

We thank the reviewer for the comment and apologize for the confusion. We changed the sentence to: "The canonical isoform was stained with both antibodies (Fig. 4a, c). The coding sequence of the novel isoform overlaps with the canonical one; thus, it cannot be specifically detected in wild-type (WT) IHCs." (Line 345-347)

5. Line 365-367, "The ABR threshold reports the function of OHCs that support the sensitivity of auditory sensing. Thus, the *Otof-ΔC* mice had normal OHC function": To evaluate the function of OHCs, DPOAE should be tested on these mice.

Response:

We thank the reviewer for noting this issue for us. We have added the DPOAE data in Supplementary figure 4a, showing that the *Otof-ΔC* mice had normal OHC function.

6. Figure 6:

- a) B, E: Please indicate the percentage of the input used in the input gel.
- b) C: The author did not specify which value the fluorescent signal is normalized to.

Response:

Thanks a lot for your suggestion. We have added the percentage of the input in the figure.

For figure 6b (revised version), the fluorescence intensity was normalized against a dark background surrounding the cell body in the same image and z-section as the previous study⁹. This is now noted in the figure legend.

7. Line 447, "except for the N terminal C2A region, the structure of the shorter alternatively spliced isoform superimposes well on the structure of the conventional one...": The two isoforms are identical with the exception of the N-terminal addition. The Alphafold2 analysis in this case thus is trivial and not very informative and could

be removed.

Response:

Thank you for your kind suggestion. According to your suggestion, we have removed the content.

8. Line 463~467, “C2B-F has the ability to bind to under different Ca²⁺ concentrations”:

This sentence is confusing. Please consider rephrasing it.

Response:

Thank you for the suggestion. The text is now being rephrased by a native speaker.

"C₂B-F has the ability to bind to Ca²⁺. Next, we investigated the effects of calcium on binding. Under different Ca²⁺ concentrations, no significant change was found in binding affinity between otoferlin (or its fragments) and endophilin A1 (Supplementary Fig. 9a-d)". (Line 471-474)

9. Line 471~473, “Collectively, these results suggested ... biological functional difference.”

There is no significant reduction of endophilin A1 colocalization in OTOF-deltaC mice in figure 6C, which weakens the conclusion that there indeed is a biological functional difference.

Response:

We appreciate the reviewer's comment. We agree this part of the manuscript was not clear enough and led to ambiguity. Figure 6C (revised Fig. 6b) here was to show that there were co-localizations between endophilin A1 in WT and *Otof-ΔC* IHCs, which indicate that they could form a complex in vivo (not to show the lower binding ability of in *Otof-ΔC* mice).

Figure 6a was an exogenous Co-IP assay performed in HEK-293T (a tool cell line from the kidney), and Figure 6C (revised Fig. 6b) was a totally in vitro assay. Considering that the otoferlin expression in the cochlea was tissue-specific; thus, immunofluorescence of IHCs Figure 6C (revised Fig. 6b) was performed to confirm

that they could really form a complex in a cochlea cell model.

We did not use immunofluorescence to define the reduction of endophilin A1 colocalization in *Otof-ΔC* mainly because the colocalization rate might be subject to bias from manual operation error, fluorescence signal intensity and resolution ratio, selection of confocal layer, etc. Furthermore, the colocalization rate between OTOF-C (or OTOF-S) and endophilin A1 does not equal to their binding ability.

For a clearer understanding, we revised the interpretation of Figure 6C (revised Fig. 6b). It reads as follows: "Taking into consideration the tissue-specificity expression of otoferlin, we performed immunofluorescence staining of IHCs. We found that endophilin A1 can colocalize with otoferlin in WT and *Otof-ΔC* mice, indicating they form an in vivo complex." (Line 460-464)

10. Figure 7:

- a) A: Scale bar. Legend for the blue channel. Remove the yellow "Merge" legend.
- b) C, G, J: Statistical analysis should be conducted on the number of organs or the animals (especially for panel G).

Response:

Thank you. According to your suggestion, we have revised Fig 7 and the figure legend.

6. In general, for the different gene transcripts, what is the average abundance of novel transcript of each gene? This information might be challenging to obtain, but if there is a way, it would be very helpful to identify the (presumably more abundant) isoforms with functional importance.

Response:

We fully agree with the reviewer that obtaining the average abundance of the novel transcript of each gene is meaningful. True biological insight into alternative splicing can indeed be found from high-quality RNA-seq data; however, accurate discovery and quantitative profiling of AS of a single cell or a gene is technically and computationally challenging currently^{10,11}. In this study, although we were not able to measure the

abundance of the novel transcript of each gene, we hope that this issue will be solved and will lead to a better understanding of splicing regulation, cell-to-cell variation, and the importance of alternative splicing in defining cell fate in the future^{12,13}.

7. Is it possible, using the SQANTI classification, to distinguish between transcripts that are generated by alternative start sites vs alternative splicing events? This will help to answer if these transcripts are regulated at the gene expression level or splicing level.

Response:

SQANTI is an automated pipeline for classifying the long-read transcripts to novel and known transcripts according to comparing their junctions to a reference annotation¹⁴. However, it cannot distinguish the transcripts generated by alternative start sites or splicing events. As the methodology improves, we may re-visit this part of the data in the future.

8. The paragraph about the *Actb* isoforms (lines 218-227) is difficult to understand. Please rephrase.

Response:

We thank the referee for this suggestion. It has been rewritten by a native speaker who is familiar with auditory research. It reads as follows: "As an example, we selected *Actb*, a ubiquitous gene expressed in all clusters (Fig. 2f), demonstrating the diversity of transcripts in each cell type (Fig. 2g, h). From short-read sequencing data, the expression level of *Actb* was indistinguishable from one cell cluster to another. However, within transcript categories, cell specificity was quite distinct (Fig. 2g). That is, among the 100 types of novel transcripts (Supplementary Data 3), 69 were different among cell types (Fig. 2g). Next, we sought to illustrate the presence of diverse transcripts of *Actb* (Fig. 2h), which have 2 transcripts referenced in the GENCODE database (VM23). A novel exon was found to be preferentially expressed in nonsensory cells (CEC and Microglia) but was absent in other cell types, highlighting cell-type-specificity."(Line 217-226)

Reviewer #2 (Remarks to the Author):

Transcriptomic analyses have been extensively employed to investigate the transcription diversity in the cochlea. However, much less is known about the alternative splicing events in the cochlea at the transcriptomic level. In the present work, the authors combined single-cell short-read and long-read RNA sequencing techniques to identify the alternative splicing events in the cochlea. They then focused on otoferlin and showed that different Otoferlin isoforms are associated with distinct IHC functions using a knockout mouse model. This study has great significance and broad interest in the related field. Some concerns and suggestions are listed below.

Response:

We thank the referee for the positive comments towards our work, and we are particularly grateful for the criticisms and suggestions that help to improve our manuscript. We have carefully addressed all the raised issues, as discussed below in a point-to-point manner and we modified the manuscript as suggested.

Major points:

1. It's important to confirm the two Otoferlin isoforms at the protein level. Can the authors detect the two isoforms in the cochlea by performing western blot?

Response:

We thank the reviewer for this question. As suggested, we performed western blot several times using different antibodies (Response Fig.3); however, none of them worked. This is likely due to the low abundance of inner hair cells that express otoferlin, representing <0.25% of the total cells in the cochlear ¹⁵.

Response Fig.3

Response Fig.3. Western blot analysis of otoferlin of the mouse cochlea (WT and *Otof*^{-/-}) using different commercial antibodies. Western blot analysis of the expression of otoferlin in WT and *Otof*-ko (*Otof*^{-/-}) mice (up to 30 cochleae were pooled for each sample with 3 repeats); however, no distinguishable objective band was found in any group.

In the present study, we generated *Otof-ΔC* and *Otof-ΔS* mice and performed whole-mount immunostaining using two otoferlin antibodies (recognizing the protein fragment encoded by exons 1-13 or exons 7-8, respectively; Fig. 4a-d) to validate whether the novel isoform was expressed at the protein level. We found that IHCs of WT and *Otof-ΔS* mice could be double-stained with two antibodies, and a complete loss of staining was observed in *Otof*^{-/-} mice. Whereas, IHCs of *Otof-ΔC* mice were stained with only one antibody, and the relative otoferlin expression level was significantly decreased (Supplementary Fig. 4c, d). Moreover, the expression of otoferlin in *Otof-ΔS* mice was also significantly decreased (Supplementary Fig. 4c, d). **Thus, these results conclude the existence of the otoferlin novel isoform at the protein level.**

2. The authors deleted the exon 2 of Otoferlin gene and obtained knockout mice that only express the novel isoform but lack the canonical isoform. The knockout mice have normal ABR threshold but reduced wave I amplitude. Further investigation revealed that the knockout mice show reduced capability of sustained exocytosis. These results are very interesting and helpful for understanding the role of otoferlin in the ribbon synapse. However, to investigate the role of the novel Otoferlin isoform (OTOFS), a more straightforward strategy would be deleting exon 6b and establishing knockout mice without functional OTOFS.

Response:

We thank the referee for this critical question and agree with you. As you suggested, we generated *Otof-ΔS* mice with deleted exon 6B depicted in the methods section. (Line 867-871)

To further explore whether knockout of the novel isoform could induce hearing impairment, we performed hearing tests on *Otof-ΔS* mice, and no ABR threshold elevation was found (**Fig. 4e, f**). We further measured the amplitude and latency of ABR wave I and no significant changes were found in *Otof-ΔS* mice (**Supplementary Fig. 5e-f**). Thus, we conclude that IHCs carry only the canonical isoforms support the normal auditory function. This notion was supported by the whole cell patch clamp recordings in *Otof-ΔS* IHCs, which showed normal Ca^{2+} current and exocytosis (**Supplementary Fig. 7e, f**).

The present study found that the novel isoform functionally overlapped the canonical isoforms but could not fully support normal vesicle recycling. We can therefore infer that the existence of the novel isoform **may be a compensatory mechanism** for maintaining normal auditory functions.

Minor points:

1. Line 47: "... without no noticeable..." should be "... and no noticeable...".

Response:

In the revised version, this sentence is removed.

2. Line 321: The authors declared that novel Otoferlin transcript is "a novel inner ear-specific transcript". It will be nice if they can validate this tissue-specific pattern by performing RT-PCR in different mouse tissues.

Response:

Thank you for your suggestion. We followed your comment and performed RT-PCR in tissues including the inner ear, cerebral cortex, cerebellum, hippocampus, and cochlear nucleus (Supplementary Fig. 3d), and found that novel otoferlin transcript was only expressed in the inner ear.

3. Line 635: The sentence "The coexisting isoforms cannot..." is confusing to the reviewer. I suggest using "the novel isoform" instead of "the coexisting isoforms".

Response:

This sentence has been revised as suggested. Thank you.

4. The language needs to be improved.

Response:

Response: We thank the reviewer for the suggestion. The manuscript has been extensively edited by a native English speaker (see acknowledgment).

Reviewer #3 (Remarks to the Author):

In this manuscript, Liu et al. used a combination of single cell- short- and long-read sequencing to explore the transcriptomic landscape of the cochlea. They complemented their study with a mass spectrometry-based proteomics to confirm their findings. They also focused on one of the novel transcripts they identified for OTOF gene; a short isoform lacking the C2A domain. Their KO mouse models showed distinct functions and expression patterns tonotopically for each isoform and suggests that the canonical isoform might be involved in hidden hearing loss.

This work provides a wealth of data and valuable insights into the transcriptomic landscape of different cell types in the cochlea. This is a much-needed knowledge that will enable research in all fields of auditory sciences. The characterization of a novel short isoform for OTOF that has a distinct function is extraordinary and has direct translational consequences for the ongoing clinical trial as well as for clinical genetic testing, natural history studies and understanding of the mechanisms involved in hidden hearing loss. Nevertheless, several issues need further clarification and correction.

Response:

We would like to thank the referee for the favorable comments to our study and are grateful to the referee for the valuable suggestions to further improve the quality of our manuscript. We re-conducted the necessary experiments and revised the manuscript as suggested.

Major comments:

1. To take full advantage of the data in this work, IHC and OHC should have been placed in two distinct clusters. We do know 1-their morphology and function are different, and 2- most deafness-associated genes are expressed in these cells, so it is surprising their data were not analyzed separately to gain valuable insights into their physiology.

Response:

Response: Thank you for the comment! We performed the downstream analysis and used the 'SCTransform' function to normalize and scale the data. PCA was performed, unsupervised clustering was applied to the top 50 PCs, and the clustering

resolution was 0.2. Finally, we identified 12 clusters that distinguished IHC and OHC subtypes based on the known marker (**Fig. 1, Supplementary Fig. 1a, b**). **In our revised manuscript, we also reassigned the long-read sequencing data according to this new single-cell cluster.**

2. To investigate the role of the novel transcript Otof-S, an indirect method was used by creating a KO mouse model for the canonical isoform Otof-C. Is there a reason why a KO of otof-S was not made instead or in addition? I think creating a KO for the otof-S is more valuable and will confirm the role of both the canonical and short isoforms.

Response:

We thank the referee for this critical question and agree with you. As you suggested, we generated *Otof-ΔS* mice with deleted exon 6B depicted in the methods section. (Line 867-871)

To further explore whether knockout of the novel isoform could induce hearing impairment, we performed hearing tests on *Otof-ΔS* mice, and no ABR threshold elevation was found (**Fig. 4e, f**). We further measured the amplitude and latency of ABR wave I and no significant changes were found in *Otof-ΔS* mice (**Supplementary Fig. 5e-f**). Thus, we conclude that IHCs carry only the canonical isoforms support the normal auditory function. This notion was supported by the whole cell patch clamp recordings in *Otof-ΔS* IHCs, which showed normal Ca^{2+} current and exocytosis (**Supplementary Fig. 7e, f**).

The present study found that the novel isoform functionally overlapped the canonical isoforms but could not fully support normal vesicle recycling. We can therefore infer that the existence of the novel isoform **may be a compensatory mechanism** for maintaining normal auditory functions.

3-A previous work by Ranum et al., (Ranum PT, Goodwin AT, Yoshimura H, Kolbe DL, Walls WD, Koh JY, He DZZ, Smith RJH. Insights into the Biology of Hearing and Deafness Revealed by Single-Cell RNA Sequencing. Cell Rep. 2019 Mar

12;26(11):3160-3171.e3. doi: 10.1016/j.celrep.2019.02.053. PMID: 30865901; PMCID: PMC6424336.) used a very similar approach combining short- and long-read sequencing to draw a picture of the inner ear transcriptome. However, there was no mention or reference to that work in this manuscript. Please reference the Ranum et al., paper and discuss your findings as compared to theirs.

Response:

We apologize for the omission of this information, which has now been added in the Discussion section in the updated version of the manuscript. It reads as follows: "Previous work by Ranum et al. was the first to explore the inner ear transcriptome by using a low-throughput approach combining short- (Smart-seq2) and long-read (Nanopore) sequencing with handpicking strategies. While they only focused on the full-length transcriptome of OHCs in the inner ear, several cell types, such as IHCs and DCs were missing in their work. In our study, we integrated short-read (Illumina) and long-read (PacBio) sequencing high-throughput methods, and the transcriptomic landscape of 12 cell types in the inner ear was explored. However, the gene numbers per cell were lower than previous work due to the insufficient data volume, and the methodology still needs to be improved in the future." (Line 558-567)

4-The wealth of data provided by this work is hard to assess using the files included in the manuscript as the comprehensive aspect is lost. I strongly believe, this data need to be made available on a website similar to what was done by Ranum et al., (<https://morlscrnaseq.org/> and https://morlscrnaseq.org/JBrowse-1.12.3/?loc=5%3A142902388..142907483&tracks=mm10_gff%2CIHC_bigwig%2CIHC_SashimiPlot&highlight=) where users could easily access and visualize the different transcripts and data supporting them. Another way to make this data accessible to the greater community is to share it on the gEAR portal (<https://umgear.org/>) or to add it as a track on the genome browser (<https://genome.ucsc.edu/>).

Response:

Thank you for reminding us. We shared our scRNA-seq data on the gEAR portal (https://umgear.org/index.html?multigene_plots=0&layout_id=0d6584b2&gene_symb

ol_exact_match=1&gene_symbol=Ocm) and added a "UCSC Genes track" file in our revised manuscript to display how to view **novel isoforms** in a specific cell type (Response Fig.2).

Response Fig.2

Response Fig.2 Method for viewing the novel isoform. Users can upload the UCSC Genes track file (Supplementary Data 7) in the UCSC Genome Browser (<https://genome-asia.ucsc.edu/>). Once uploaded, custom track data can be manipulated to view the novel isoforms in different cell types.

5-Several gene therapies have been developed for OTOF-related hearing loss and a clinical trial is ongoing. The relevance of this work and its translational aspect need to be discussed. would the short isoform alone rescue hearing? Is there a need for dual vectors use for gene therapy for OTOF? Etc...

Response :

Thank you for reminding us. This is truly an important issue to discuss. Following your suggestion, we discussed this issue in the revised manuscript (Line 581-593).

6-The number of cells isolated for each cell type as well as the methodology used to assign them to a specific cluster are not reported. This is very relevant to the

interpretation of the data and needs to be added.

Response:

Thanks for this suggestion. We now annotated the number of cells in each cell type in Figure 1.

Following the reviewer's suggestions, we also reannotated cell clusters to cell types according to the known markers in our results section (Line 162-169, **Supplementary Fig. 1a, b**).

Minor Comments:

1-The manuscript needs to be reviewed for conciseness (Some sentences are 5-6 lines long). It will also benefit from a revision of syntax and grammar in some sections.

Response:

The manuscript has been extensively edited by a native English speaker. We believe it is easy to follow now.

2-It would be great to add a summary figure to show the proposed tonotopic expression and distinct roles of Otof-C and Otof-S in IHC.

Response:

We apologize for confusing the reviewer in the original version of the manuscript. The two isoforms did not possess distinct roles but instead are functionally overlapped.

In our opinion, adding a summary figure to illustrate the different expressions of the two otoferlin isoforms is unnecessary. Because in this study, we only observed that the expression ratio of the two-isoform exhibited tonotopic differences, the molecular mechanisms accounting for this phenomenon remain to be explored in future studies.

3-Figures: Please change figures' color palette to make them color-blind friendly

Response:

Thank you for the suggestion. According to your suggestion, we have changed the figures' color palette in our revised manuscript.

4-Fig2B and 3F: please add the abbreviations for isoform type to the pie chart itself. It makes it easier to understand.

Response:

Thank you for the suggestion. To make it easier to follow, we added color legends on the side of each figure in our revised manuscript.

5-Fig2F and 3G: please plot cell types on the pie chart. Avoid putting numbers inside the pie charts.

Response:

Thank you for the suggestion. We added color legends on the side of each figure in our revised manuscript.

6-Line 365: "The ABR threshold reports the function of OHCs..." This is not accurate. DPOAE report the function of OHC. ABR threshold reports both IHC and OHC.

Response:

You are right that IHC also participated. We added the DPOAE results in **Supplementary figure 4a**, prove that the *Otof-ΔC* mice had normal OHC function.

7-Please make sure to provide more specifics and explain some of your statements when needed. For example: Line363 "This was distinct from previous studies in other otoferlin mutation models"

Response:

Thank you for reminding us. We have revised this sentence "This was distinctly different from previous studies in other otoferlin mutation models that exhibited profound hearing loss caused by depletion of overall otoferlin levels." (Line 361-363)

8-Lines 295-297: is there a table associated with this data?

Response:

Thank you for reminding us. We have added it (Supplementary Data 6).

9-Lines 286-292: Where there peptides identified in your MS-based proteomics that were absent in your ScISOr-Seq isoforms? Please indicate and discuss.

Response:

Good question! Although this is in our opinion beyond the scope of present study, we did perform mass spectrometry (MS)-based proteomics to determine whether the ScISOr-Seq isoforms were translated into proteins.

To address reviewer's comment, we reanalyzed the data and found about 1000 peptides were absent in our ScISOr-Seq isoforms. Two reasons may account for this discrepancy: First, we used the organ of Corti from P26 – 30 wild-type mice, and the cell types or their proportions may be different from our ScISOr-Seq. Second, the limitation of long-read sequencing is its moderate sequencing depth, which fall short of accurate quantification of genes with less abundant mRNAs, such as *Apob*, *Nefh*, *Mog*, *Slc17a7*, *Tuba3b*, et al.

10-Line 231: change 5'/3'-donor to 5'/3'-donor or acceptor.

Response:

T Thank you and revised as suggested.

11-Line 299: "In addition, 81 ScISOr-Seq specific novel peptides were discovered that were absent from the UniProt database." It will be great to add a table listing these novel peptides and corresponding proteins/genes and what makes them novel.

Response:

Thank you for reminding us. We have added it (Supplementary Data 7).

Reference

- 1 Michalski, N. *et al.* Otoferlin acts as a Ca(2+) sensor for vesicle fusion and vesicle pool replenishment at auditory hair cell ribbon synapses. *Elife* **6**, doi:10.7554/eLife.31013 (2017).
- 2 Strenzke, N. *et al.* Hair cell synaptic dysfunction, auditory fatigue and thermal sensitivity in otoferlin Ile515Thr mutants. *Embo j* **35**, 2519-2535, doi:10.15252/embj.201694564 (2016).
- 3 Roux, I. *et al.* Otoferlin, defective in a human deafness form, is essential for exocytosis at the auditory ribbon synapse. *Cell* **127**, 277-289, doi:10.1016/j.cell.2006.08.040 (2006).
- 4 Hernandez, V. H. *et al.* Optogenetic stimulation of the auditory pathway. *J Clin Invest* **124**, 1114-1129, doi:10.1172/JCI69050 (2014).
- 5 Thorpe, R. K. *et al.* The natural history of OTOF-related auditory neuropathy spectrum disorders: a multicenter study. *Hum Genet* **141**, 853-863, doi:10.1007/s00439-021-02340-w (2022).
- 6 Schwander, M. *et al.* A forward genetics screen in mice identifies recessive deafness traits and reveals that pejvakin is essential for outer hair cell function. *J Neurosci* **27**, 2163-2175, doi:10.1523/jneurosci.4975-06.2007 (2007).
- 7 Kroll, J. *et al.* Endophilin-A regulates presynaptic Ca(2+) influx and synaptic vesicle recycling in auditory hair cells. *Embo j* **38**, doi:10.15252/embj.2018100116 (2019).
- 8 Duncker, S. V. *et al.* Otoferlin couples to clathrin-mediated endocytosis in mature cochlear inner hair cells. *J Neurosci* **33**, 9508-9519, doi:10.1523/jneurosci.5689-12.2013 (2013).
- 9 Li, S. *et al.* Myosin-VIIa is expressed in multiple isoforms and essential for tensioning the hair cell mechanotransduction complex. *Nat Commun* **11**, 2066, doi:10.1038/s41467-020-15936-z (2020).
- 10 Wright, C. J., Smith, C. W. J. & Jiggins, C. D. Alternative splicing as a source of phenotypic diversity. *Nat Rev Genet* **23**, 697-710, doi:10.1038/s41576-022-00514-4 (2022).
- 11 Buen Abad Najar, C. F., Yosef, N. & Lareau, L. F. Coverage-dependent bias creates the appearance of binary splicing in single cells. *Elife* **9**, doi:10.7554/eLife.54603 (2020).
- 12 Gupta, I. *et al.* Single-cell isoform RNA sequencing characterizes isoforms in thousands of cerebellar cells. *Nat Biotechnol*, doi:10.1038/nbt.4259 (2018).
- 13 De Paoli-Iseppi, R., Gleeson, J. & Clark, M. B. Isoform Age - Splice Isoform Profiling Using Long-Read Technologies. *Front Mol Biosci* **8**, 711733, doi:10.3389/fmolb.2021.711733 (2021).
- 14 Tardaguila, M. *et al.* SQANTI: extensive characterization of long-read transcript sequences for quality control in full-length transcriptome identification and quantification. *Genome Res* **28**, 396-411, doi:10.1101/gr.222976.117 (2018).
- 15 Ehret, G. & Frankenreiter, M. Quantitative analysis of cochlear structures in the house mouse in relation to mechanisms of acoustical information processing. *Journal of comparative physiology* **122**, 65-85 (1977).

REVIEWERS' COMMENTS

Reviewer #1 (Remarks to the Author):

Liu et al. have addressed most of the suggestions that we raised in the previous review. Notably, the authors have generated a mouse model in which the OTOF-S isoform is specifically deleted, which has strengthened the completeness of the study. Also, in the point-by-point response and discussion section, the authors have addressed our questions regarding the discrepancies observed in the phenotype of OTOF isoform-specific deletion mice compared to previous literature. Based on the phenotype of the two isoforms, the authors significantly changed their conclusions, and now state that the two isoforms have rather overlapping functions, which is more consistent with the presented data. Additionally, the revised version of the manuscript has shown significant improvement in statistics, figure quality, and readability.

Concerns still remain:

- despite Reviewer 3's suggestion to divide inner hair cells and outer hair cells into two distinct clusters, the PacBio UMAP only shows a single hair cell cluster (HC). This raises the question of whether this outcome is a result of limitations in the ScISOr-Seq data analysis. Overall, there are still limitations to using ScISOr-Seq with inner ear samples.

- it is overall puzzling why certain genes have such outlandish high numbers of transcripts: 100 for beta-actin, and even more for Col2a1. It is hard to envision 100 biologically significant splice forms of actin, which is one of the most sequence-conserved proteins. How much of that is transcriptional noise?

- How many of the newly discovered transcripts contain a premature stop codon? This could inform on whether such transcripts give rise to function proteins, or whether they are transcriptional noise, or are part of a protein level regulation mechanism, similar to what was reported by Drummond and Friderici in 2013 (where incorporation of a novel exon into gamma actin creates a premature stop, thus helps to downregulate protein levels).

- A very limited sampling of known hair cell genes (e.g. Myo15, XIRP2, etc) failed to identify known published isoforms, which makes the reviewer suspect that the lack of depth of Iso-seq is a significant limiting factor. This might be limitations of the PacBio platform, but it would be important to set the expectation level of reader. Outlining the limits of this study would therefore be helpful.

Minor language issues:

- line 121-123: Sentence needs to be reworded. Maybe something along the lines of:

- o We generated an otoferlin canonical isoform (OTOF-C) deletion (OTOF-122 Δ C) animal model, revealing a and demonstrated that this novel transcript could be translated into a functional protein isoform that functionally overlaps with the canonical one and decreases vesicle recycling.

- Line 125: In addition, compared with the canonical isoform, the novel otoferlin isoform should exhibited decreased binding potential with synaptic proteins such as endophilin A1

Overall, if the reviewers provide satisfactory answers to the questions above, this manuscript should meet the requirements for publication.

Reviewer #2 (Remarks to the Author):

The authors have addressed most of my comments and greatly improved their manuscript. I have only one last suggestion. Several splicing regulators (such as RBM24 and SRRM4) have been shown to be highly expressed in the hair cells and regulate hair cell-specific alternative splicing. It will be helpful for the readers if the authors can discuss this in their manuscript.

Reviewer #3 (Remarks to the Author):

It is evident that the authors have put in a considerable amount of effort to improve the quality and clarity of the manuscript. I appreciate their diligence in revising the manuscript, and I believe that the changes made have enhanced the value of the work.

I have just one minor comment that I believe would further improve the accuracy of the manuscript: In the section discussing FDA-approved gene therapies for OTOF-related hearing loss, I recommend adding "Akouos" to the list of companies that have been granted FDA approval for their OTOF gene therapy.

Reviewer #4 (Remarks to the Author):

The revised version is improved and this study provides a transcriptomic resource that could be of great utility to the inner-ear research community.

My expertise is in bioinformatics and I can comment on this aspect – the computational data analysis is well performed. Sharing the data through the gEAR portal is very important.

The authors did not directly address the question I raised (regarding quantitative analysis of the novel transcripts). But they referred to this point in their response to other referees. I accept that such analysis is beyond the scope of this exploratory study and is left for future investigations.

Point-by-Point Response to NCOMMS-22-36208A

Reviewer #1 (Remarks to the Author):

Liu et al. have addressed most of the suggestions that we raised in the previous review. Notably, the authors have generated a mouse model in which the OTOF-S isoform is specifically deleted, which has strengthened the completeness of the study. Also, in the point-by-point response and discussion section, the authors have addressed our questions regarding the discrepancies observed in the phenotype of OTOF isoform-specific deletion mice compared to previous literature. Based on the phenotype of the two isoforms, the authors significantly changed their conclusions, and now state that the two isoforms have rather overlapping functions, which is more consistent with the presented data.

Additionally, the revised version of the manuscript has shown significant improvement in statistics, figure quality, and readability.

Response:

We express our gratitude to the referee for the constructive review.

Concerns still remain:

- despite Reviewer 3's suggestion to divide inner hair cells and outer hair cells into two distinct clusters, the PacBio UMAP only shows a single hair cell cluster (HC). This raises the question of whether this outcome is a result of limitations in the ScISOr-Seq data analysis. Overall, there are still limitations to using ScISOr-Seq with inner ear samples.

Response: We appreciate the reviewer's comment.

As we mentioned in our manuscript, the key challenge was the relatively low sequencing depth of gene number per cell of the long-read data (Line 183-190). So, it's hard to profile cellular diversity similar to the scRNA-seq; however, it can identify the major cell types. This and the integration analysis of the scRNA-seq and Iso-seq data show our sequence data's high quality and accuracy in the two platforms.

Using scRNA-seq to profile cellular diversity and cell barcodes as a guide to

allocate long reads to each cell type of the cochlea can compensate for this limitation.

¹ The PacBio UMAP only shows a single hair cell cluster, while we can still divide it into inner hair cells and outer hair cells according to the cell barcodes of scRNA-seq data. Thus, we believe ScISOr-Seq is suitable for inner ear samples.

We have now included in the discussion about the limitation of the current method (Line 653-658).

- it is overall puzzling why certain genes have such outlandish high numbers of transcripts: 100 for beta-actin, and even more for Col2a1. It is hard to envision 100 biologically significant splice forms of actin, which is one of the most sequence-conserved proteins. How much of that is transcriptional noise?

Response: We appreciate the reviewer's comment.

Transcription and translation of genes were a complex and intricate process ². For the ScISOr-Seq, quantitative analysis of the novel transcripts is still hard to achieve due to the technical and computational challenges. And without quantitative analysis of the novel transcripts, it is hard to say which isoform was abundant and whether these isoforms are transcriptional noises.

To validate these novel isoforms, we used bulk RNA-seq and Sanger sequencing, and we observed that more than 45.31% of the novel isoforms could be verified, which expanded our understanding of transcriptome diversity.

In response to the reviewer's comment, we have expanded our discussion regarding this issue: "Our research expanded our understanding of transcriptome diversity in the cochlear; however, we cannot exclude the presence of potential transcriptional noise, which awaits future studies to verify." (Line 579-581).

- How many of the newly discovered transcripts contain a premature stop codon? This could inform on whether such transcripts give rise to function proteins, or whether they are transcriptional noise, or are part of a protein level regulation mechanism, similar to what was reported by Drummond and Friderici in 2013 (where incorporation of a novel exon into gamma actin creates a premature stop, thus helps to downregulate

protein levels).

Response:

We fully agree with the reviewer; however, it is very difficult to count how many transcripts contain a premature stop codon and this is beyond the scope of this exploratory study.

"predicted_NMD", one of the outputs from the software "SQANTI" in "Supplementary data 3", **maybe an alternative** to predict whether the novel isoform could translate a functional protein.

In the example such as the novel transcript of *Calb1* (PB.21971.7, Figure 3e), we identified a novel exon (exon 5b), which is similar to exon 3a of *Actg1* (Drummond and Friderici in 2013). The "predicted_NMD" of this novel transcript is "true", indicating that exon 5b contains a premature stop codon and is more likely to be degraded by nonsense-mediated decay (NMD). Moreover, we found that about 71.4% of the novel transcripts, which have the coding potential, might be translated into proteins.

However, the functions of these transcripts remain to be explored.

- A very limited sampling of known hair cell genes (e.g. *Myo15*, *XIRP2*, etc) failed to identify known published isoforms, which makes the reviewer suspect that the lack of depth of Iso-seq is a significant limiting factor. This might be limitations of the PacBio platform, but it would be important to set the expectation level of reader. Outlining the limits of this study would therefore be helpful.

Response:

In response to the reviewer's comment, **we have included in our discussion:** "Current limitations in long-read sequencing are found in its moderate sequencing depth compared to short-read sequencing. This impedes the accuracy of identifying and quantifying the less abundant mRNAs, such as *Myo15*, *Xirp2*, etc., and future investigations still needed to overcome this limitation. Thus, our Iso-seq experiments mainly focused on discovering uncharacterized transcript isoforms rather than their expression levels." (Line 653-658)

Minor language issues:

- line 121-123: Sentence needs to be reworded. Maybe something along the lines of:

- We generated an otoferlin canonical isoform (OTOF-C) deletion (OTOF-122 Δ C) animal model, revealing and demonstrated that this novel transcript could be translated into a functional protein isoform that functionally overlaps with the canonical one and decreases vesicle recycling.

- Line 125: In addition, compared with the canonical isoform, the novel otoferlin isoform should exhibited decreased binding potential with synaptic proteins such as endophilin A1.

Response:

This sentence has been revised as follow. Thank you. (Line 119-124)

Overall, if the reviewers provide satisfactory answers to the questions above, this manuscript should meet the requirements for publication.

Reviewer #2 (Remarks to the Author):

The authors have addressed most of my comments and greatly improved their manuscript. I have only one last suggestion. Several splicing regulators (such as RBM24 and SRRM4) have been shown to be highly expressed in the hair cells and regulate hair cell-specific alternative splicing. It will be helpful for the readers if the authors can discuss this in their manuscript.

Response:

We express our gratitude to the referee for the constructive review process. Your new comments have been incorporated into the revised manuscript. (Line 573-575)

Reviewer #3 (Remarks to the Author):

It is evident that the authors have put in a considerable amount of effort to improve the quality and clarity of the manuscript. I appreciate their diligence in revising the manuscript, and I believe that the changes made have enhanced the value of the work.

I have just one minor comment that I believe would further improve the accuracy of the manuscript: In the section discussing FDA-approved gene therapies for OTOF-related hearing loss, I recommend adding "Akouos" to the list of companies that have been granted FDA approval for their OTOF gene therapy.

Response: Thank you very much for your approval of our revised work. As suggested, we have added company name Akouos. (Line 589)

Reviewer #4 (Remarks to the Author):

The revised version is improved and this study provides a transcriptomic resource that could be of great utility to the inner-ear research community.

My expertise is in bioinformatics and I can comment on this aspect – the computational data analysis is well performed. Sharing the data through the gEAR portal is very important.

The authors did not directly address the question I raised (regarding quantitative analysis of the novel transcripts). But they referred to this point in their response to other referees. I accept that such analysis is beyond the scope of this exploratory study and is left for future investigations.

Response: We sincerely thank the reviewer for the positive view of our revised manuscript.

Reference

1. Gupta I, *et al.* Single-cell isoform RNA sequencing characterizes isoforms in thousands of cerebellar cells. *Nat Biotechnol*, (2018).
2. Webster MW, Weixlbaumer A. The intricate relationship between transcription and translation. *Proc Natl Acad Sci U S A* **118**, (2021).